# A secreted effector with a dual role as a toxin and as a transcriptional factor

Dandan Wang [1,5], Lingfang Zhu [1,5], Xiangkai Zhen[2,5], Daoyan Yang [1], Changfu Li [1], Yating Chen[1], Huannan Wang[2], Yichen Qu [1], Xiaozhen Liu[1], Yanling Yin [3], Huawei Gu[1], Lei Xu [1], Chuanxing Wan[3], Yao Wang [1], Songying Ouyang [2] ✉ & Xihui Shen [1,4] ✉

Bacteria have evolved multiple secretion systems for delivering effector proteins into the cytosol of neighboring cells, but the roles of many of these effectors remain unknown. Here, we show that *Yersinia pseudotuberculosis* secretes an effector, CccR, that can act both as a toxin and as a transcriptional factor. The effector is secreted by a type VI secretion system (T6SS) and can enter nearby cells of the same species and other species (such as *Escherichia coli*) via cell-cell contact and in a contact-independent manner. CccR contains an N-terminal FIC domain and a C-terminal DNA-binding domain. In *Y. pseudotuberculosis* cells, CccR inhibits its own expression by binding through its DNA-binding domain to the *cccR* promoter, and affects the expression of other genes through unclear mechanisms. In *E. coli* cells, the FIC domain of CccR AMPylates the cell division protein FtsZ, inducing cell filamentation and growth arrest. Thus, our results indicate that CccR has a dual role, modulating gene expression in neighboring cells of the same species, and inhibiting the growth of competitors.

Chemical communication is vital for the coordination of the activities and behaviors of multicellular and unicellular species[1–4]. All chemical communications require signal molecules and their cognate receptors to transform intercellular messages into intracellular responses. The chemical nature of signals is diverse, including lipids, peptides, proteins, and polysaccharides[4–9]. The receptors are either cell-surface transmembrane receptors that bind to signals that do not penetrate the cell to activate downstream intracellular signaling cascades, or intracellular ligand-regulated transcription factors that perceive chemical signals capable of penetrating cells to control gene expression[4,5,8].

In bacteria, the best-characterized chemical communication process is quorum sensing, which involves the release of and response to small signal molecules termed autoinducers[10–12]. Although bacteria have evolved at least seven types of protein secretion systems (T1SS-T7SS) for delivering intracellularly produced proteins into the extracellular milieu or into the cytosol of neighboring eukaryotic or prokaryotic cells[13,14], the role of these secreted proteins in signaling transduction remains largely unknown. T6SS is a well-defined bacterial weapon for injecting toxic effectors into neighboring competitor cells and protecting kin cells with immunity proteins[15–18]. While T6SS is traditionally recognized as a contact-dependent bacterial weapon for

[1]State Key Laboratory of Crop Stress Biology for Arid Areas, Shaanxi Key Laboratory of Agricultural and Environmental Microbiology, College of Life Sciences, Northwest A&F University, Yangling, Shaanxi 712100, China. [2]The Key Laboratory of Innate Immune Biology of Fujian Province, Provincial University Key Laboratory of Cellular Stress Response and Metabolic Regulation, Biomedical Research Center of South China, Key Laboratory of OptoElectronic Science and Technology for Medicine of the Ministry of Education, College of Life Sciences, Fujian Normal University, Fuzhou 350117, China. [3]Xinjiang Production and Construction Corps Key Laboratory of Protection and Utilization of Biological Resources in Tarim Basin, College of Life Science, Tarim University, Alar 843300 Xinjiang, China. [4]College of Plant Protection, Northwest A&F University, Yangling, Shaanxi 712100, China. [5]These authors contributed equally: Dandan Wang, Lingfang Zhu, Xiangkai Zhen. ✉e-mail: ouyangsy@fjnu.edu.cn; xihuishen@nwsuaf.edu.cn

microbe-host and microbial interspecies competition[15,17–20], a contact-independent, receptor-dependent T6SS killing pathway has also been reported[21], involving the release of a microcin-like effector with cell-entry properties, thus allowing T6SS attack from a distance. Intriguingly, in a population of bacteria, a large amount of produced toxin is likely delivered to "self" cells rather than to competitor cells, which seems to be a waste of toxin. One proposed hypothesis to explain this phenomenon is that these self-exchanged toxins perform an interbacterial signaling function to promote cooperative behaviors[15].

The widespread FIC (filamentation induced by cAMP) proteins are characterized by the presence of a conserved nine-residue signature motif HXFX(D/E)GNGRXXR termed FIC[22]. FIC proteins regulate a variety of molecular processes in organisms ranging from bacteria to humans by catalyzing posttranslational modifications (PTM), including AMPylation[23], UMPylation[24], phosphorylcholination[25], and phosphorylation[26]. The PTM activity of FIC was originally discovered in effector proteins delivered by the T3SS or T4SS secretion systems of some pathogenic bacteria to manipulate host cell processes[23–25]. Later, PTM activity was also identified in the toxin component of bacterial toxin–antitoxin modules (e.g., Doc-Phd[26], FicT-FicA[27] and VbhT-VbhA[28]) and FICD/HYPE[29] in eukaryotic cells that regulate the unfolded protein response.

In this study, we identify a T6SS-secreted bifunctional FIC protein (hereafter referred to as CccR) that mediates interspecies bacterial competition by AMPylation of the cell division protein FtsZ in nonself cells, and mediates cell-to-cell communication by acting as a transcriptional regulator in surrounding kin cells.

## Results

### CccR mediates interbacterial antagonism

Previously, we identified a gene locus (YPK_0952-0958) encoding multiple T6SS effector-immunity pairs in the *Yersinia pseudotuberculosis* (*Yptb*) genome. Among these effectors, YPK_0954 (Tce1) was characterized as a nuclease toxin that mediates contact-independent T6SS antagonism[21]. Remarkably, an open reading frame (YPK_0951, hereafter referred to as CccR) containing an FIC motif was identified upstream of this locus (Fig. 1a). Notably, the carboxyl terminus of CccR harbors a helix-turn-helix (HTH) DNA-binding domain, which sets it apart from all other functionally characterized FIC proteins (Fig. 1b). Inducible expression of CccR in *Escherichia coli* resulted in significant growth inhibition, and the inhibition was dependent on the FIC motif, as a mutation in the conserved His$_{192}$, which is critical for the catalysis of adenylylation, eliminated the inhibition (Fig. 1c). We also examined the cell morphology by microscopic analysis and found that *E. coli* cells expressing CccR became filamentous. In contrast, similar to the control strain containing empty vector, no filamentous cells were found for strains expressing CccR$^{H192A}$ (Fig. 1d and Supplementary Fig. 1a–c). These results show that CccR exerts FIC toxicity when expressed in *E. coli*.

Next, we overexpressed CccR in *Yptb* with the pME6032 vector to verify whether it is a T6SS effector. Overexpressing CccR in *Yptb* did not result in growth arrest and cell filamentation (Supplementary Fig. 2a, b), suggesting the existence of an unknown immunity mechanism that protects the producing cells from intoxication. Although the secretion of CccR was readily detected in *Yptb* supernatant, it was completely abolished in Δ*clpV1-4*, a mutant defective in all four sets of T6SSs, indicating that CccR is a genuine T6SS effector (Supplementary Fig. 2c). The secretion of CccR was greatly reduced in the Δ*clpV3* and Δ*clpV4* mutants but not in the Δ*clpV1* and Δ*clpV2* mutants (Supplementary Fig. 2c), and the secretion defect of Δ*clpV1-4* was markedly restored by complementation of *clpV3* and *clpV4* (Supplementary Fig. 2d), together indicating that CccR is secreted by T6SS-3 and T6SS-4.

Having demonstrated that CccR is a T6SS-secreted toxin, we next sought to investigate whether it participates in bacterial interspecies

antagonism. To this end, we performed interspecies competition assays between relevant *Yptb* strains and *E. coli* cocultured under contact-promoting conditions or in liquid medium. Under contact-promoting conditions, the *Yptb* WT donor exhibited a 3-fold growth advantage in competition with the *E. coli* recipient (Fig. 1e). Unexpectedly, the *Yptb* WT donor exhibited an even stronger competitive advantage (5-fold) over the *E. coli* recipient in liquid medium (Fig. 1f). Under both conditions, the growth advantage was dramatically

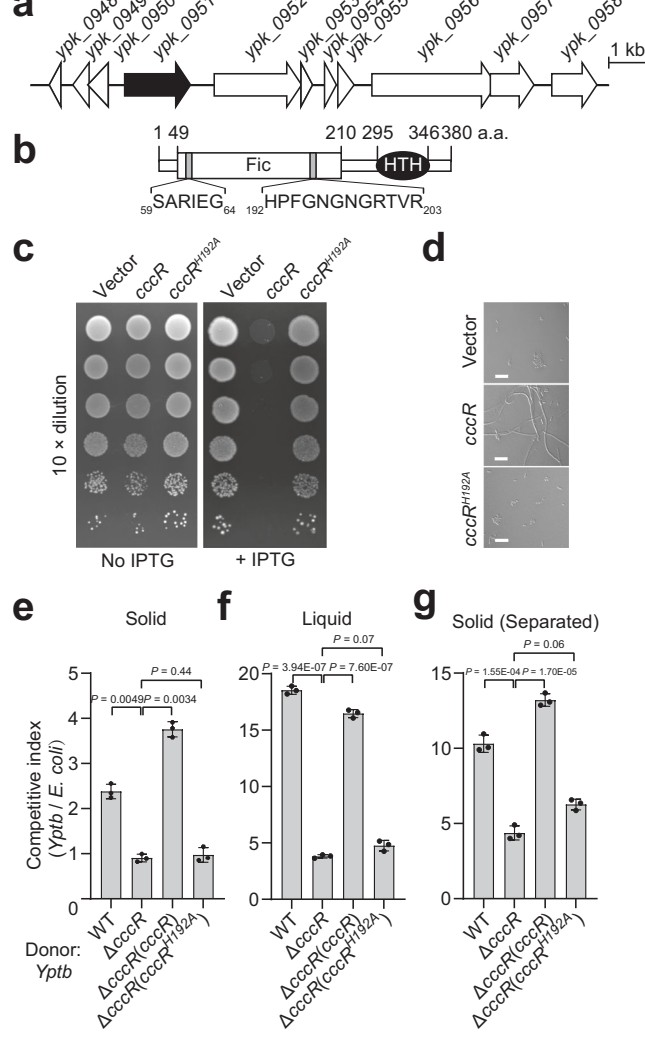

**Fig. 1 | CccR mediates both contact-dependent and contact-independent competition. a** Schematic of a gene cluster encoding T6SS effectors in *Yptb*. Locus tag numbers are provided on the top of each gene, and the *cccR* gene (*ypk_0951*) is indicated in black. **b** Domain organization of CccR. The boundaries for the FIC (residues 49–210) and HTH (residues 295–346) domains are indicated. **c** CccR is toxic to *E. coli*. Growth of *E. coli* BL21(DE3) cells containing a vector control or a vector expressing CccR or CccR$^{H192A}$ under noninducing (no IPTG) or inducing (100 μM IPTG) conditions. **d** CccR induces filamentation of *E. coli* cells. Representative micrographs of *E. coli* BL21(DE3) cells expressing CccR or CccR$^{H192A}$ were acquired 10 h after induction of protein expression. Scale bar, 10 μm. The images shown in **c** and **d** are representative of three separate experiments with similar results. **e–g** Outcome of growth competition between the indicated *Yptb* donor strains and *E. coli* DH5α on solid support (**e**), in liquid medium (**f**) or by separating donor and recipient cells with a cell-impermeable membrane on the surface of solid medium (**g**). The CFU ratio of the donor and recipient strains was measured based on plate counts. Data in **e–g** are presented as the mean ± standard deviation (SD) of three independent experiments. *P*-values from all data were determined using a two-sided, unpaired Student's *t*-test, and differences were considered significant at *P* < 0.05. Source data are provided as a Source Data file.

abrogated by deletion of *cccR* from the WT donor. Moreover, the fitness defect of the Δ*cccR* mutant was completely restored by complementation of WT *cccR* but not the catalytically inactive *cccR^H192A* allele (Fig. 1e, f). These results suggest that CccR, similar to Tce1, mediates not only contact-dependent but also contact-independent T6SS antagonism. The role of CccR in the mediation of contact-independent antagonism was further confirmed by repeating the competition assays on the surface of solid medium with a cell-impermeable membrane to separate the donor and recipient cells (Fig. 1g).

Consistent with its role in the mediation of contact-independent antagonism, addition of the CccR protein to the liquid medium led to complete growth arrest and filamentation of *E. coli* cell. In contrast, no such phenomena were observed after the addition of the CccR^H192A or heat-inactivated CccR protein (Supplementary Fig. 3a–d). These results suggest that, unlike canonical T6SS effectors, CccR possesses a T6SS needle-independent cell-entry mechanism that allows it to enter target cells possibly through binding to transporters on the surface of target cells in a manner similar to that of bacteriocins.

The finding that CccR mediates both contact-dependent and contact-independent interbacterial antagonism in vitro prompted us to further investigate whether it can facilitate the overcoming of colonization resistance by the enteropathogen *Yptb* through antagonism of commensal *E. coli*. To this end, mice treated with streptomycin were colonized with *E. coli* DH5α for 24 h before being challenged with *Yptb* WT or Δ*cccR*. After 36 h of *Yptb* challenge, the *E. coli* intestinal load of mice challenged with *Yptb* WT was significantly lower than that of mice challenged with Δ*cccR* (Supplementary Fig. 4a). In contrast, *Yptb* WT exhibited significantly higher levels of colonization in mice precolonized with *E. coli* relative to Δ*cccR* (Supplementary Fig. 4b).

## CccR engages FhuA for cell entry

To identify the putative receptors required for CccR cell entry, we performed a GST pull-down screening assay by incubation of GST-CccR-coated beads with total lysates of *E. coli*. Proteins specifically retained by GST-CccR were detected through silver staining after sodium dodecyl sulfate–polyacrylamide gel electrophoresis (SDS–PAGE). Mass spectrometric analysis identified 2 proteins with similar molecular weights in the 40-kDa band: the cell division protein FtsZ (B21_00095) and elongation factor-Tu (EF-Tu, B21_03141). In addition, a weak band at 80 kDa (which became clear visible upon prolonging the staining duration) was identified as FhuA, a TonB-dependent siderophore receptor (B21_00148) (Fig. 2a). Interestingly, FhuA has been well-established as an outer membrane transporter for the cell entry of bacteriocins such as colicin M[30] and microcin 25[31]. The interaction between CccR and FhuA was confirmed by bacterial two-hybrid, with significant β-galactosidase activity observed in the *E. coli* BTH101 strain coexpressing CccR and FhuA fused to the T18 and T25 fragments of adenylate cyclase (CyaA), respectively (Supplementary Fig. 5a). Moreover, FhuA protein could be copurified with GST-CccR in the in vitro binding assay, further validating the direct binding between CccR and FhuA (Fig. 2b). To map the binding domain of CccR on FhuA, we constructed N-terminal fusions of GST with fragments of CccR of different lengths and examined the binding of the fusion proteins to FhuA (Supplementary Fig. 5b). The in vitro binding assay revealed that the N-terminal region of CccR (AAs 1-110) is required for binding to FhuA, further supporting the occurrence of direct interaction between CccR and FhuA.

The direct binding of CccR to FhuA prompted us to examine whether FhuA serves as a transporter for mediation of CccR entry into target cells. To test this hypothesis, we performed a fluorescence-based assay using Alexa Fluor 488-conjugated CccR to probe its import in vivo. While the addition of AF488-CccR to *E. coli* WT cells yielded fluorescent bacterial cells, the Δ*fhuA* mutant was not labeled. However,

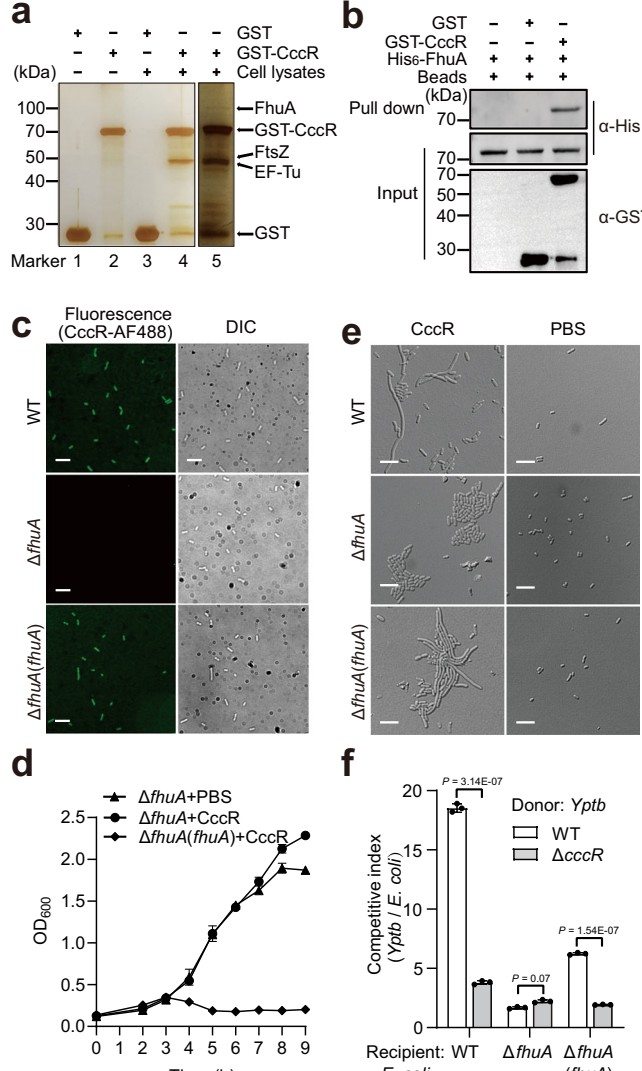

**Fig. 2 | CccR engages the outer membrane transporter FhuA for target cell entry. a** Identification of proteins specifically retained by GST-CccR from cell lysates of *E. coli* BL21(DE3) with a silver-stained SDS–PAGE gel. The identities of the retained proteins were identified through mass spectrometry. **b** Direct interaction between CccR and FhuA verified with an in vitro GST pull-down assay. His₆-FhuA was incubated with GST-CccR or GST, and the protein complexes captured on glutathione beads were detected using western blotting. The blots shown in **a** and **b** are representative of three independent experiments with similar results. **c** Cell entry of CccR-AF488 into the indicated *E. coli* strains examined by fluorescence microscopy. Note that the Δ*fhuA* mutant showed no labeling, while *E. coli* WT and the complemented strain Δ*fhuA(fhuA)* were labeled (scale bar, 10 μm). **d** Growth curves of Δ*fhuA* and Δ*fhuA(fhuA)* treated by exogenous supplementation with CccR protein were obtained by measuring the OD₆₀₀ at 1 h intervals. Data are presented as the mean ± standard deviation (SD) of three independent experiments. **e** Effects of CccR in inducing cell filamentation. Representative micrographs of the indicated *E. coli* strains treated with exogenously supplemented CccR protein were acquired 4 h after CccR treatment. The images shown are representative of three separate experiments with similar results. Scale bar, 10 μm. **f** Outcome of growth competition between the indicated *Yptb* donor strains and *E. coli* recipient strains in liquid medium. The CFU ratio of the donor and recipient strains was measured based on plate counts. Data in **e** and **f** are presented as the mean ± standard deviation (SD) of three independent experiments. *P*-values from all data were determined using a two-sided, unpaired Student's *t*-test, and differences were considered significant at *P* < 0.05. Source data are provided as a Source Data file.

the fluorescence defect of Δ*fhuA* was completely restored to the WT level by complementation (Fig. 2c). These results confirmed that CccR enters target cells through binding to the outer membrane transporter FhuA. Consistent with the observation that the N-terminal region of CccR is required for binding to FhuA, the AF488-conjugated CccR$_{101-380}$ failed to enter *E. coli* WT cells (Supplementary Fig. 5c). The fact that the N-terminal portion of CccR is important for internalization into prey cells further supports that this toxin behaves in a colicin-like manner in terms of prey cell entry.

If FhuA does play essential roles in facilitating CccR entry into target cells, we predict that the *E. coli* Δ*fhuA* mutant must show natural resistance to CccR treatment. As predicted, deletion of *fhuA* abolished the growth arrest effect of exogenously supplied CccR protein to *E. coli* WT. In contrast, similar to *E. coli* WT, the growth of the Δ*fhuA*(*fhuA*) complemented strain was severely inhibited by the addition of the CccR protein to the medium (Fig. 2d). Consistent with these findings, while addition of CccR protein induced filamentation of *E. coli* WT and Δ*fhuA*(*fhuA*) complemented strain cells, the cell shape of Δ*fhuA* was not affected (Fig. 2e and Supplementary Fig. 5d).

To further investigate whether FhuA is involved in CccR-mediated contact-independent T6SS antagonism, we performed interspecies competition assays in liquid medium. As shown in Fig. 2f, while the *Yptb* WT donor showed a CccR-dependent competitive advantage over the *E. coli* WT recipient, the CccR-dependent competitive advantage was diminished when the strain was cocultured with the *E. coli* Δ*fhuA* recipient. Notably, the CccR-dependent competitive advantage of the *Yptb* WT donor was restored when the strain was cocultured with the Δ*fhuA*(*fhuA*) complemented strain. These results indicate that the secreted CccR adopts a bacteriocin-like cell-entry mechanism to compete with target cells.

## CccR targets FtsZ to induce filamentation

While EF-Tu has been reported to be a physiologically relevant target of FIC toxins[26], the cell division machinery has never been reported as an FIC target, although this protein has been referred to as the filamentation induced by cAMP protein since its discovery in 1982[32]. To determine whether the essential cell division factor FtsZ is a physiologically relevant target of CccR, we first verified the specific interaction between CccR and FtsZ (from both *E. coli* and *Yptb*) with in vitro binding and bacterial two-hybrid assays (Fig. 3a and Supplementary Fig. 6a–c). To further determine whether CccR could modify FtsZ by AMPylation, we coexpressed CccR and FtsZ in *E. coli* BL21 tagged with glutathione S-transferase (GST) and His$_6$, respectively. Purified His$_6$-FtsZ was separated by SDS–PAGE, excised from the gel, and digested with trypsin. The resulting peptides were analyzed by liquid chromatography coupled with tandem mass spectrometry (LC–MS/MS). An additional peptide with a mass shift of 329 Da corresponding to AMPylation was identified in His$_6$-FtsZ coexpressed with wild-type CccR but not the CccR$^{H192A}$ variant. Sequencing revealed that the peptide was -FEPMELT$^8$NDAVIK- with AMPylation at position Thr$^8$ (Fig. 3b). The MS results were confirmed with in vitro AMPylation assays with bio-17-ATP as the substrate. Incubation of purified GST-CccR with His$_6$-FtsZ and bio-17-ATP led to robust AMPylation of FtsZ. No AMPylation signal was detected when His$_6$-FtsZ$^{T8A}$ was used, confirming that Thr$^8$ of FtsZ is the modification site (Fig. 3c). Consistent with the observation that a mutation in the conserved histidine residue abolished its toxicity to bacterial cells, GST-CccR$^{H192A}$ failed to catalyze the reaction, indicating the importance of the FIC motif in this enzymatic activity (Fig. 3c). These results indicated that CccR functions as an AMP transferase that targets FtsZ at Thr$^8$. Sequence alignment analysis showed that the Thr$^8$ residue of FtsZ is conserved in Enterobacteriaceae but not in other families (Supplementary Fig. 6d). Consistently, in vitro AMPylation assays showed that CccR can modify FtsZs from *Yptb* and *Salmonella enterica*, suggesting it cannot distinguish FtsZ of *Yptb* from that of other bacteria such as *E. coli* and

*S. enterica* (Supplementary Fig. 6e). It will be interesting to determine whether this FtsZ-targeting mechanism is conserved among Enterobacteriaceae species that contain the conserved Thr$^8$ residue.

In most bacteria, cytokinesis is initiated by the localization of the essential protein FtsZ, a GTPase and tubulin homolog that can form ring-like structures and generate constrictive forces. Hydrolysis of GTP bound to FtsZ protofilaments is thought to drive a straight-to-curved conformational change and generate the constrictive force required for cell division[33,34]. To determine the effects of CccR AMPylation on the activity of FtsZ, we determined the GTPase activity of FtsZ that had been pretreated with CccR in the presence of ATP. Whereas untreated FtsZ or FtsZ that had been pretreated with CccR$^{H192A}$ showed readily detectable GTP hydrolysis, the GTPase activity of FtsZ was reduced in a dose-dependent manner after pretreatment with increasing amounts of CccR (Fig. 3d).

Next, we determined the effects of CccR on the polymerization of FtsZ by a sedimentation assay. FtsZ monomers were incubated with different concentrations of His$_6$-CccR in the presence of ATP and GTP. The FtsZ monomer and polymers in the reactions were separated by ultracentrifugation and analyzed by SDS–PAGE. When FtsZ monomers were incubated with increasing amounts of CccR, the amounts of FtsZ polymers in the pellets decreased in a dose-dependent manner (Fig. 3e). Note that the catalytically inactive CccR$^{H192A}$ mutant had no effect on FtsZ polymerization. These results demonstrate that CccR inhibits FtsZ polymerization.

The effect of CccR on inhibition of FtsZ polymerization was further verified by negative stain transmission electron microscopy (TEM) observations. As shown in Fig. 3f, in the absence of CccR, purified FtsZ formed long linear polymers (~1,000 nm in length), indicating that it had polymerized successfully in a GTP-dependent manner. Incubation with CccR but not the CccR$^{H192A}$ variant completely abolished FtsZ polymerization. The finding that CccR inhibits FtsZ filament formation does not rule out the possibility that CccR also disassembles or destabilizes existing filaments. However, when we treated preformed FtsZ filaments with the same amount of CccR in the presence of ATP, we did not observe changes in FtsZ polymerization. These results show that CccR affects the polymerization of FtsZ but not the depolymerization of FtsZ filaments.

In exponentially growing cells, FtsZ localizes predominantly to the mid-cell region. In a GTP-dependent fashion, dynamic filaments of the protein assemble to form a characteristic ring, referred to as the Z ring, which templates the cell division machinery, including peptidoglycan biosynthetic enzymes[33–35]. To observe the effect of CccR modification on Z ring formation with fluorescence microscopy, we used cells coding for a functional FtsZ variant containing mNeonGreen inserted at a permissive site instead of the wild-type *ftsZ* allele (FtsZ-FP)[35,36]. Whereas the *E. coli* cells harboring the plasmid control or those expressing the CccR$^{H192A}$ variant showed an apparent fluorescent band of assembled FtsZ-FP at the cell septa, the fluorescent band disassembled into a collection of randomly distributed puncta in the filamentous CccR$^{WT}$-expressing cells (Fig. 3g). Similar results were obtained when the CccR$^{WT}$, CccR$^{H192A}$ or heat-treated CccR$^{WT}$ protein was added to FtsZ-FP-labeled *E. coli* cells (Fig. 3h). Taken together, these results indicated that the essential cell division factor FtsZ is a physiologically relevant target of CccR and that CccR AMPylates FtsZ to inhibit Z ring formation and further disrupts cell division, leading to cell filamentation.

## Crystal structure of CccR

To understand the biological role of CccR at the molecular level, we determined its crystal structure using the single-wavelength anomalous method (SAD) at a 2.78 Å resolution. The crystallographic data collection and structure refinement statistics are listed in detail in Supplementary Table 1. The crystallographic asymmetric unit contains two molecules of CccR assembled into a butterfly-shaped homodimer.

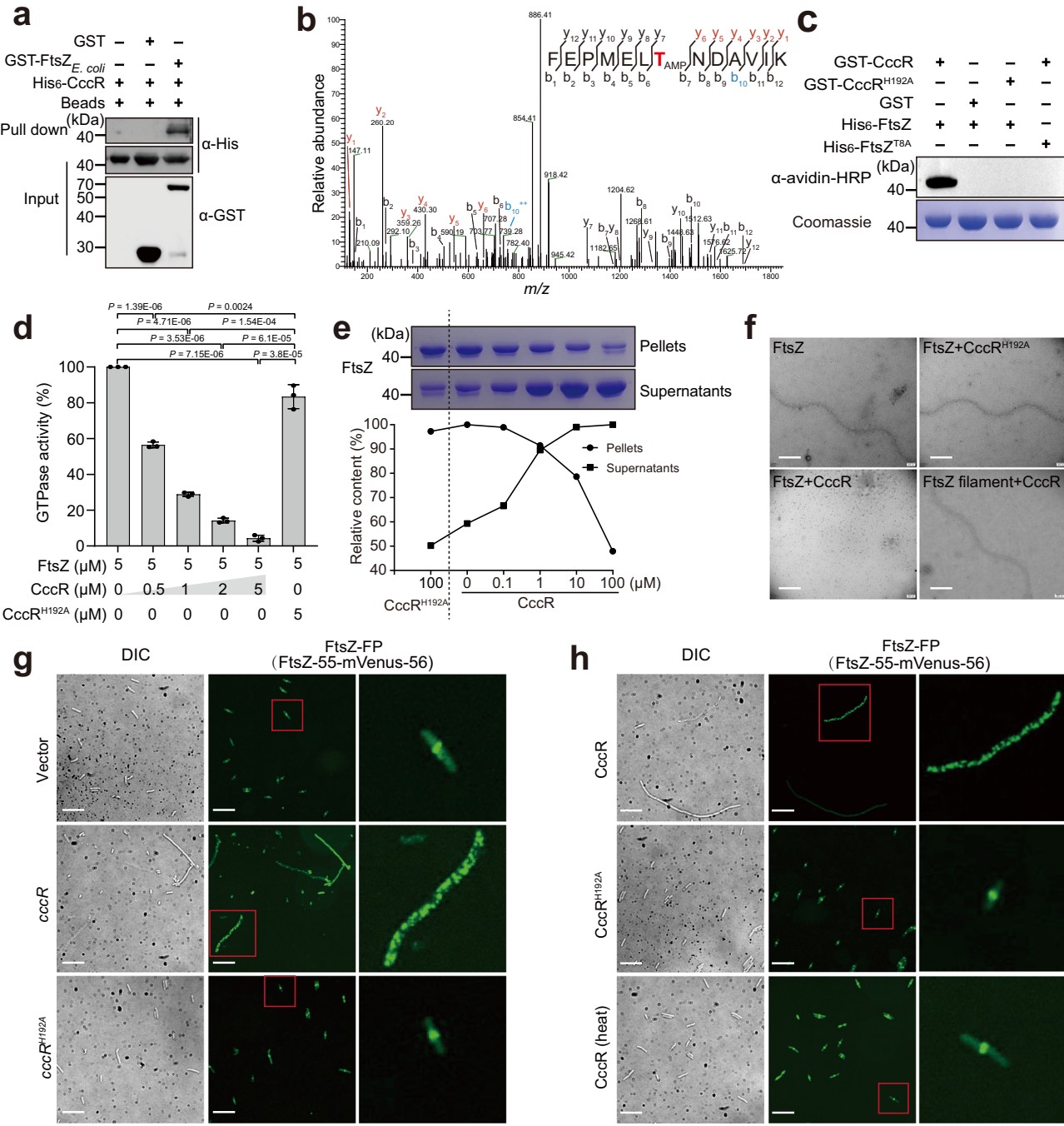

**Fig. 3 | CccR AMPylates FtsZ to inhibit cell division. a** Direct interaction between CccR and FtsZ verified with an in vitro GST pull-down assay. The blots shown are representative of three independent experiments with similar results. **b**–**c** CccR AMPylates FtsZ on Thr[8]. **b** Tandem mass spectrum of the indicated peptide from His$_6$-FtsZ purified from *E. coli* cells coexpressing GST-CccR and His$_6$-FtsZ. Fragmentation ions (b, blue; y, red) with resolved spectra and the site of AMPylation (red) are indicated. **c** Representative blot using avidin-HRP to detect biotinylated proteins following incubation of FtsZ proteins with bio-17-ATP and CccR proteins. Coomassie bright blue staining is shown as a loading control. The blots shown are representative of three independent experiments with similar results. **d** CccR treatment abolished the GTPase activity of FtsZ in a dose-dependent manner. The data are the mean ± SD from three biological replicates. *P*-values from all data were determined using a two-sided, unpaired Student's *t*-test, and differences were considered significant at *P* < 0.05. **e** CccR inhibits FtsZ polymerization. FtsZ polymerization in the presence of CccR was allowed to proceed for 60 min, followed by ultracentrifugation at 100,000 × g for 40 min. Upper panel: The resulting supernatants and pellets were analyzed by SDS–PAGE followed by Coomassie brilliant blue staining. Lower panel: Quantification of the percentage of polymerized FtsZ versus total FtsZ. The band intensity was quantified with ImageJ. The blots shown are representative of three independent experiments with similar results. **f** Negative stain electron microscopy analysis of the effects of CccR on FtsZ polymerization by incubation of purified FtsZ with CccR or CccR[H192A] and analysis of the effects of CccR on depolymerization of preformed FtsZ filaments by incubation of FtsZ filaments with CccR. Scale bar, 100 nm. **g**–**h** Effects of CccR on Z ring formation. Fluorescence microscopy observation of *E. coli* cells expressing FtsZ-FP and carrying plasmids for the inducible expression of CccR and CccR[H192A] (**g**) or treated with exogenously provided CccR, CccR[H192A] or heat inactivated CccR (**h**). A magnified view of bacteria in the red box is shown in the right column. The images shown in **f**–**h** are representative of three separate experiments with similar results. Scale bar, 10 μm. Source data are provided as a Source Data file.

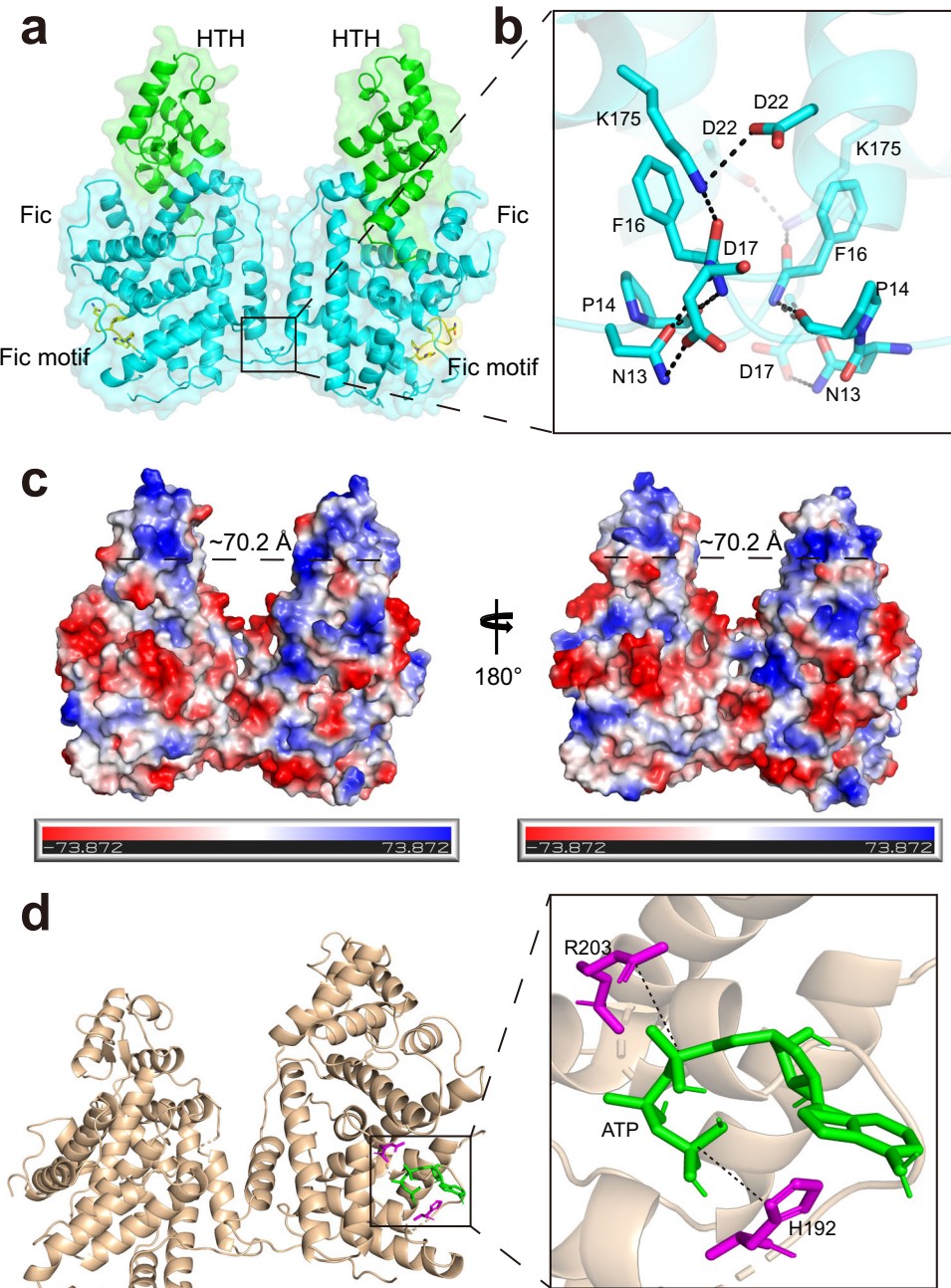

**Fig. 4 | Overall structure of CccR. a** Cartoon mode of the structure of CccR. Ribbon mode of the overall structure of CccR. The FIC domain contains an α helical structure (residues 1–80, purple), an active center structure (residues 81–275, blue) and an HTH domain (residues 276–380, green). **b** Interaction analysis of the dimeric interfaces between the two monomers of CccR. The residues involved in the interactions are shown as sticks. **c** Electrostatic surface potential of CccR. The CccR features of positively charged patches; blue and red indicate positive and negative charges, respectively, and the distance between them in the dimer is indicated. **d** Molecular docking of CccR with ATP. The conformation with the lowest docking energy was determined using Chimera software. Key residues of CccR involved in ATP binding are shown as purple sticks, and ATP is shown as green sticks. Two potential hydrogen bonds are indicated by dashed lines.

Homodimerization was also confirmed using size-exclusion chromatography (SEC) and analytical ultracentrifugation analysis (AUC) (Fig. 4a and Supplementary Fig. 7a). The two CccR molecules within the homodimer are nearly identical to each other, as they were superimposed with a root-mean-square deviation value of 0.593 Å over 340 Cα atoms. Despite low sequence identity shared with the typical FIC family protein SO_4266 (gi|24375750, PDB:3eqx, 29% with 45% coverage)[37], CccR adopts a classic FIC protein family fold consisting of two domains, namely, the N-terminal FIC domain containing the conserved FIC motif and the C-terminal HTH-type DNA-binding domain with a large positively charged region (Fig. 4a–c and Supplementary

Fig. 7b, c). The dimerization interface area is less than 800 Å², accounting for only 4% of the monomer surface area. Dimerization is mainly mediated by interactions formed between the α1 helices and the N-terminal loops of each CccR monomer. Specifically, the side chain of residue K175 from the α5-α6 loop of one monomer forms a salt bridge with the side chain of residue D22 from the α1 helix of the other monomer. The remainder of the interactions are formed between the residues on the N-loops of the two monomers: the side chain of residue N13 hydrogen bonds with the side chain of D17 from the other monomer, and concurrently, the main chains of N13 and P14 of one monomer hydrogen bond with the main chain of F16 of the other

monomer (Fig. 4b). AUC analysis indicated that simultaneous substitution of N13 and K175 with alanine abolishes dimerization (Supplementary Fig. 4d). Strikingly, compared to dimeric FIC (PDB: 3eqx), the DNA-binding domains of CccR are separated by back-to-back association of the CccR monomers, which has not been reported for other FIC proteins. The distance between the α12 helices within each DNA-binding domain is approximately 70.2 Å, suggesting that CccR recognizes and binds a particular DNA operator (Fig. 4c and Supplementary Fig. 7c).

FIC proteins perform adenylylation, which is dependent on the small-molecule nucleotide substrate ATP and transfer AMP groups; the FIC motif HXFX(D/E)GNGRXXR is important for recognition and catalysis of the substrate. Molecular docking showed that the active center of CccR bound ATP, and $H_{192}$ and $R_{203}$ in the FIC motif ($H_{192}$PFGNGNGRTVR$_{203}$) formed hydrogen bonds with ATP (Fig. 4d). The conserved $H_{192}$ residue plays a key role in catalysis and its imidazole group deprotonates and activates the hydroxyl group for nucleophilic attack of the high-energy pyrophosphate bonds of the nucleotide substrate. The $R_{203}$ residue in the FIC motif could bind the ribose ring and γ-phosphate of nucleotides through the guanidine group, thus playing a role in the recognition of small-molecule nucleotide substrates[38,39]. As a family of toxic proteins, most FIC proteins are expressed in an inhibited form autoinhibited by a conserved motif (S/T)XXXE(G/N) termed the inhibitory α-helix ($α_{inh}$), which obstructs the active site to prevent optimal positioning of the ATP substrate for AMP transfer. Based on the location of the $α_{inh}$, FIC proteins were grouped into three classes: the $α_{inh}$ located on an interacting protein reminiscent of toxin-antitoxin modules (Class I), or located N- terminal (Class II) or C- terminal (Class III) to the FIC motif within the same polypeptide chain. As a Class II FIC protein, CccR contains an N terminal $α_{inh}$ ($S_{59}$ARIEG). The crystal structure of CccR shows $S_{59}$ and $E_{63}$ form a stable hydrogen bond with $R_{203}$ to prevent binding of the ATP substrate for modification (Supplementary Fig. 7e).

## CccR acts as a transcriptional regulator

Toxin-antitoxin modules usually bind to their own promoters to autoregulate transcription[40,41]. To investigate whether CccR acts as a transcriptional regulator to autoregulate *cccR* expression, we measured the expression of CccR by using a chromosomal $P_{cccR}$::*lacZ* transcriptional fusion introduced into the chromosomes of *Yptb* WT and the Δ*cccR* mutant. As shown in Fig. 5a, the LacZ activity of the Δ*cccR* mutant was significantly increased and exceeded that of the *Yptb* WT by a factor of 3.3, and this increase could be substantially restored to the WT level by introducing a complementary plasmid expressing CccR. Next, we detected the binding of CccR to its promoter region with an electrophoresis mobility shift assay (EMSA). Incubation of CccR$_{201-380}$, but not CccR$_{1-200}$, with the $P_{cccR}$ promoter probe (−271 to −1 bp relative to the ATG initiation codon of the *cccR* gene) led to retarded mobility of the probe in a dose-dependent manner. However, no probe shifts were observed for the negative controls (Supplementary Fig. 8a, b). These results demonstrated that CccR, serving as a repressor, autoregulates *cccR* expression by directly binding to its promoter region. To further identify the precise binding sites for CccR in the promoter region, DNase I footprinting analysis was performed. Footprinting analyses showed that CccR protects a 46-nt sequence (AGATAATCAT-N$_{26}$-ATGATAATCT) from DNase I degradation; this region is composed of two inverted repeats of the 10-bp AGATAATCAT sequence separated by 26 bp (Fig. 5b and Supplementary Fig. 8a). The 26-bp spacer between binding boxes is unusually large but is compatible with the 70.2 Å separation between DNA-binding helices observed in the CccR apo structure. A similar phenomenon was observed for the *Bacillus subtilis* SPbeta phage transcriptional regulator AimR, in which the DNA-binding helices were separated by 75 Å and recognized an operator with a 25-bp spacer[42]. We hypothesized that the palindrome sequence plays a crucial role in

DNA binding and transcriptional repression. As predicted, upon mutation of the palindromes in the $P_{cccR}$ promoter, binding of CccR$_{201-380}$ to the $P_{cccRm}$ promoter probe was abolished in the EMSA (Supplementary Fig. 8b). Next, we constructed vectors to express CccR under the control of its native promoter $P_{cccR}$ or the palindrome mutated $P_{cccRm}$ promoter in *E. coli*, and the expression of CccR was examined by western blotting. While the expression of CccR was rather low under the control of its native promoter $P_{cccR}$, mutation of the palindromes drastically increased CccR expression (Fig. 5c).

We also employed isothermal titration calorimetry (ITC) to quantify the interactions between the CccR variants and the $P_{cccR}$ promoter probe (Supplementary Fig. 8c–f). The 100-bp $P_{cccR}$ promoter probe containing both palindrome sequences was synthesized and titrated into CccR, CccR$_{1-220}$ and CccR$_{201-380}$. The binding affinity ($K_D$) of the $P_{cccR}$ promoter for CccR was 2.56 ± 0.12 µmol/L, with an overall stoichiometry of 1:2. While the binding capacity of the $P_{cccR}$ promoter for CccR was comparable to that for the N-terminus deleted variant CccR$_{201-380}$ ($K_D$ = 6.13 ± 0.11 µmol/L), it was 250-fold stronger than that for the C-terminal HTH domain deleted variant CccR$_{1-220}$ ($K_D$ = 1.53 ± 0.13 mmol/L), further verifying the transcriptional regulator activity of the C-terminal HTH domain. Notably, the binding capacity of CccR for the palindrome mutated $P_{cccRm}$ promoter was completely abrogated, supporting the specific binding of CccR to the promoter. Together, these results demonstrated that the FIC family toxin CccR acts as a transcriptional regulator to negatively autoregulate *cccR* expression by a feedback mechanism through binding to the palindrome sequences near the −35 and −10 transcriptional boxes in the $P_{cccR}$ promoter. To further investigate whether dimerization is required for the regulatory activity of CccR, we evaluated the effect of the CccR$^{N13A/K175A}$ mutant on the repression activity of the $P_{cccR}$ promoter. As shown in Fig. 5a, complementation with the wild-type *cccR* gene, but not the *cccR*$^{N13A/K175A}$ mutant gene, restored the repression effect on the $P_{cccR}$ promoter in the Δ*cccR* mutant. In addition, the binding affinity of the CccR$^{N13A/K175A}$ mutant to the palindromic binding site was reduced by a factor of 148 (Supplementary Fig. 9a). These results are consistent with the prediction that CccR, as a transcriptional regulator, must be present as a dimer to recognize the palindromic binding site. We also examined the toxicity of the CccR$^{N13A/K175A}$ mutant by expressing it in *E. coli*; the mutation had marginal effects on its toxicity and AMPylation activity (Supplementary Fig. 9b, c). These data reinforce the idea that the regulatory and toxic activities of CccR are independent.

Given that CccR is a T6SS effector, we wondered whether the transcriptional regulator activity of CccR is still functional after it is delivered into neighboring cells. To test this hypothesis, we added purified CccR protein into the medium of the Δ*cccR* reporter strain containing the $P_{cccR}$::*lacZ* transcriptional fusion. Strikingly, LacZ activity was significantly inhibited by the CccR protein in a dose-dependent manner, suggesting that CccR acts as a transcriptional regulator not only in the producing cells but also in the target cells in which it is delivered (Fig. 5d). To further explore this phenomenon under natural conditions, we employed the Transwell assay by coincubation of the recipient reporter strain Δ*cccR*($P_{cccR}$::*lacZ*) with various donor strains in wells separated by a 0.4-µm-pore-size polyester membrane, and the LacZ activity in the recipient reporter strain was detected after coincubation for 5 h. As shown in Fig. 5e, the $P_{cccR}$::*lacZ* activity in the Δ*cccR*($P_{cccR}$::*lacZ*) recipient strain was significantly repressed when coincubated with *Yptb* WT, and this repression was dramatically alleviated by coincubation with Δ*cccR*. Remarkably, the repression was restored by coincubation with the complemented strains Δ*cccR*(*cccR*) and Δ*cccR*(*cccR*$^{H192A}$) but not with Δ*cccR*(*cccR*$_{1-220}$), which was defective in the C-terminal HTH domain. These results demonstrated that CccR delivered from neighboring bacteria can act as a functional transcriptional regulator to control gene expression in recipient cells after cell entry. Consistent with

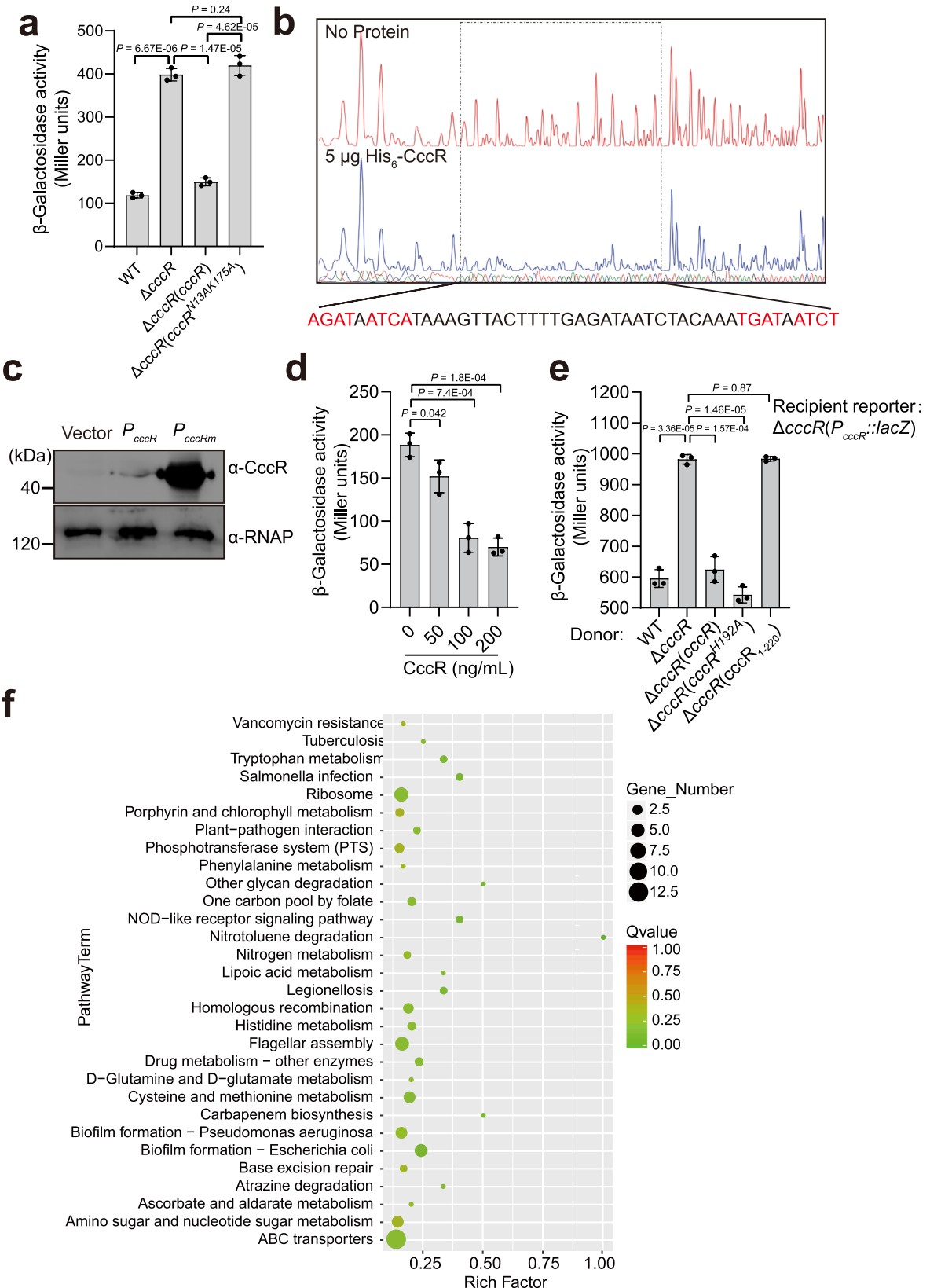

**f**

this, deletion of the CccR receptor gene, *fhuA*, in the Δ*cccR*(P*cccR*::*lacZ*) recipient abolished the repression effect of *Yptb* WT coincubated in separate wells of the Transwell system, and the repression effect was fully restored by complementation of *fhuA* in the Δ*cccR*Δ*fhuA*(P*cccR*::*lacZ*) recipient (Supplementary Fig. 10), further corroborating that CccR acts as an intercellular transcriptional

regulator to mediate cell-to-cell communication once it has entered target recipient cells.

To systematically identify genes regulated by delivered CccR in the Δ*cccR* recipient cells, we performed RNA sequencing (RNA-seq)-based comparative transcriptomic analysis of Δ*cccR* cells coincubated with *Yptb* WT or Δ*cccR* in separate wells of a Transwell system. The

**Fig. 5 | CccR regulates gene expression in both producing and recipient cells.**
**a** β-Galactosidase activity analyses of *cccR* promoter activity using the transcriptional $P_{cccR}$::*lacZ* chromosomal fusion reporter expressed in the *Yptb* WT strain, Δ*cccR* mutant strain and complemented strain Δ*cccR*(*cccR*) grown to stationary phase. Data are presented as the mean ± standard deviation (SD) of three independent experiments. *P* values from all data were determined using a two-sided, unpaired Student's *t*-test, and differences were considered significant at *P* < 0.05. **b** CccR-binding site analysis using a DNase I footprinting assay. The binding sequence is shown below, and the inverted repeat sequences are highlighted in red. **c** Expression of CccR in *E. coli* examined by western blotting. The expression of CccR is under the control of the native $P_{cccR}$ promoter or the palindrome mutated $P_{cccRm}$ promoter. The blots shown are representative of three separate experiments

with similar results. **d** Effect of exogenously provided CccR protein (0, 50, 100 and 200 ng/ml) on $P_{cccR}$ promoter activity examined with *Yptb* WT containing the $P_{cccR}$::*lacZ* chromosomal fusion reporter. **e** Effect of the CccR protein delivered from the indicated *Yptb* strains on $P_{cccR}$ promoter activity in the recipient reporter cells. Transwells were used to separate the donor and recipient strains. Data in **d** and **e** are presented as the mean ± standard deviation (SD) of three independent experiments. *P* values in **d** and **e** were determined using a two-sided, unpaired Student's *t*-test, and differences were considered significant at *P* < 0.05. **f** The KEGG enrichment bubble chart of up-regulated genes and down-regulated genes in Δ*cccR* cells co-incubated with *Yptb* WT versus co-incubated with Δ*cccR* in Transwells. Source data are provided as a Source Data file.

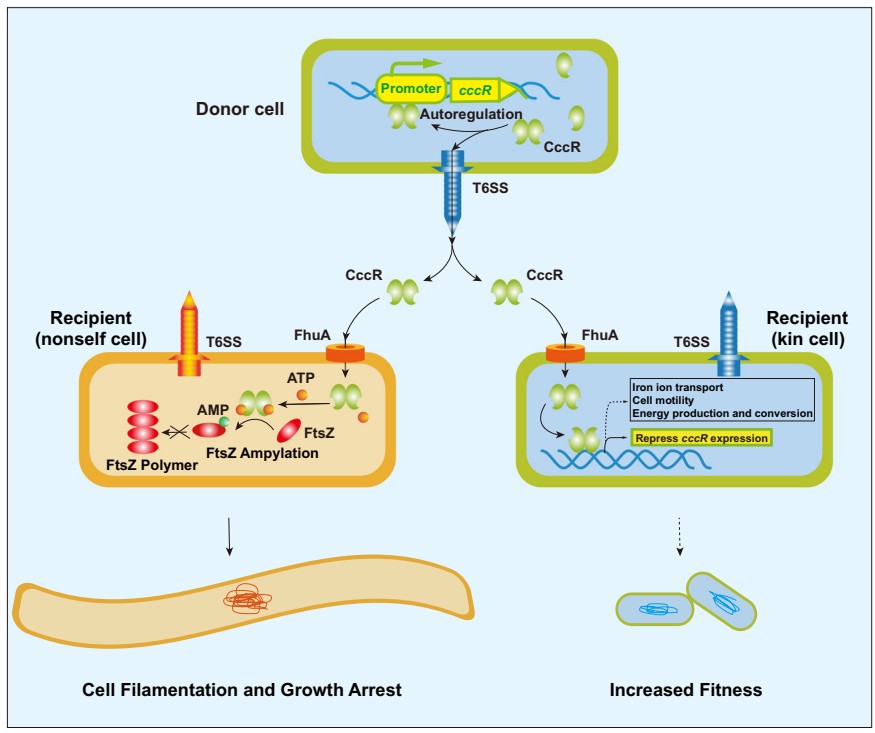

**Fig. 6 | Proposed model for the dual role of CccR in bacterial competition and cell-to-cell communication.** In CccR-producing cells, CccR represses its own expression. Under competition conditions, CccR is released into the extracellular milieu to relieve the autorepression. Released CccR enters target cells by engaging the TonB-dependent outer membrane transporter FhuA in a bacteriocin-like manner. In the presence of many prey cells (Left), secreted CccR is preferentially internalized by prey cells, where it AMPylates cell division protein FtsZ, leading to

cell filamentation and growth arrest. Following the decrease in prey cells (Right), secreted CccR is internalized by sister cells to repress *cccR* expression by acting as a transcriptional regulator, informing the population that its production is unnecessary. Under this condition, CccR may also act as a global regulator that regulates expression of genes involved in iron acquisition, motility, and energy production to coordinate bacterial behaviors and increase bacterial fitness. The potential transcriptional regulatory effects of CccR in prey cells are unclear at present.

expression of a total of 447 genes was up- or downregulated more than 1.2-fold. The transcriptomic results were validated by quantitative reverse transcriptase polymerase chain reaction (qRT-PCR) analysis of 16 representative genes by directly adding the catalytically inactive CccR[H192A] protein to Δ*cccR* mutant cells (Supplementary Fig. 11a). The functions of the differentially expressed genes (DEGs) were identified using Kyoto Encyclopedia of Genes and Genomes (KEGG) pathway analysis, and the major pathways regulated by delivered CccR are summarized in Fig. 5f and Supplementary Data 1. The DEGs were involved in multiple KEGG pathways, and the most enriched pathways were ABC transporters, ribosome synthesis, energy production and conversion, and flagellar assembly. Among ABC transporter pathway genes, 36 regulated genes were involved in inorganic ion transport and metabolism. Remarkably, 11 genes related to iron ion transport were all upregulated, while genes encoding bacterioferritin and nonheme ferritin were downregulated. Among the cell motility genes, 10 genes involved in flagellar motility were upregulated, while 5 genes involved

in fimbrial biogenesis were downregulated. In addition, 16 of 19 genes related to energy production and conversion were downregulated (Supplementary Fig. 11b, c). These data suggest that CccR delivered into kin cells may play roles in coordinating bacterial behaviors in nutrition acquisition, motility and energy production.

## Discussion
In this study, we examined CccR, a bifunctional T6SS toxin that mediates interspecies bacterial competition by AMPylation of the cell division protein FtsZ in nonself cells and mediates cell-to-cell communication by acting as a transcriptional regulator upon delivery into neighboring kin cells (Fig. 6). To the best of our knowledge, this is the first report of an intercellular transcriptional regulator that can regulate gene expression not only in its producing cells but also in neighboring cells, thus conferring the producing cells the ability to coordinate neighboring bacterial behaviors. While all chemical communications require signals and their cognate receptors to transform

intercellular messengers into intracellular responses[4,6,7], we reveal here that the delivered CccR acts as both a signal mediator capable of penetrating recipient cells and a transcriptional regulator to transform the intercellular messengers into intracellular responses directly. These findings reveal a previously undescribed one-molecule mechanism for cell-to-cell communication and highlight a role for T6SS in cell signaling beyond the well-known competition function in the microbial community.

Previously, it was reported that a contact-dependent growth inhibition (CDI) system in *Burkholderia thailandensis* delivers BcpA, a DNase toxin, that induces changes in the expression of 841 genes in immune target cells[43]. Although the molecular mechanism underlying this process, termed contact-dependent signaling (CDS), requires the catalytic activity of BcpA, the details remain unclear. In this study, we provide evidence of a role of CccR as a transcriptional regulator that regulates gene expression by directly binding DNA. RNA-seq analysis resulted in the identification of 447 genes that were up- or down-regulated more than 1.2-fold after CccR delivery. However, we cannot identify putative CccR-binding elements in the promoter regions of these regulated genes, and therefore it is not clear how CccR affects their expression. CccR might recognize promoters in an unspecific manner, or perhaps FIC-derived toxicity may induce stress that is responsible for the effect. It will be interesting to determine whether CccR can act as a transcriptional regulator in prey cells. However, given that CccR toxicity inhibits prey cell growth, and no CccR-binding elements have been identified in the *E. coli* genome, the relevance of the putative activity of CccR as transcriptional regulator in prey cells is unclear.

While the activity of Class I FIC can be directly regulated by removing the interacting antitoxin, Class II and III FIC proteins may require complex intrinsic or extrinsic factors to modulate expulsion of the inhibitory $\alpha_{inh}$[22]. For example, the toxicity of some Class III FIC proteins is regulated by oligomerization[44], autoadenylylation[45], and metal ions[46]. Although Class II FIC proteins account for 80% of total FIC proteins, nearly nothing is known about their biological functions and activity modulation. Here, we found that the Class II FIC protein CccR auto-represses its own expression by acting as a transcriptional regulator, providing a new perspective for understanding the immune mechanism of toxic proteins. However, future work is needed to investigate the identity of putative intrinsic and/or extrinsic factors that maintain the $\alpha_{inh}$ in an inhibitory state in self cells, and trigger expulsion of the $\alpha_{inh}$ to relieve autoinhibition in nonself cells.

While each antibacterial T6SS effector is often co-expressed with a cognate immunity protein encoded in a bicistron to protect the producing cells and sister cells from intoxication[15-18], no immunity protein for CccR was identified in its neighboring ORFs. We speculate that when CccR is produced, it may be bound by T6SS components (Hcp, PAAR or unknown chaperones) to prevent it from being toxic. Its repressor function could aim to prevent toxicity in cases where its binding to T6SS components is insufficient. The same is true of sister cells. In the presence of many prey cells, secreted CccR proteins will be preferentially internalized by prey cells. After the prey cells are killed, secreted CccR protein might act as a signal molecule to inform the population that its production is unnecessary, because it is mainly internalized by sister cells in these conditions. This hypothesis is partially supported by our findings that CccR exhibits significant higher affinity to *E. coli* FhuA than to *Yptb* FhuA and enters *E. coli* cells more efficiently than *Yptb* cells (Fig. 6 and Supplementary Fig. 12).

In conclusion, our findings reveal a previously undescribed one-molecule mechanism for cell-to-cell communication and highlight a role for T6SS in cell signaling beyond the well-known competition function in microbial communities. FIC proteins containing HTH domains are widely distributed in bacteria (Supplementary Fig. 13), which suggests that this one-molecule mechanism might represent a common strategy for cell-to-cell communication in bacteria.

## Methods

### Ethics statement
All mouse experimental procedures were performed in accordance with the Regulations for the Administration of Affairs Concerning Experimental Animals approved by the State Council of People's Republic of China. The protocol was approved by the Animal Welfare and Research Ethics Committee of Northwest A&F University (protocol number: NWAFUSM2018001). Six-week-old female mice (BALB/c) were purchased from the central animal laboratory of Xi'An JiaoTong University (Xi'an, China) and kept in a temperature ($24 \pm 2\,°C$), $50 \pm 10\%$ humidity, air flow of 35 exchanges and light-controlled room (12 h light, 12 h darkness) with free access to food and water.

### Bacterial strains, constructs and growth conditions
Bacterial strains and plasmids used in this study are listed in Supplementary Data 2. Primers used in this study are listed in Supplementary Data 3. *E. coli* strains were grown in LB medium (tryptone 1%, yeast extract 0.5%, NaCl 1%) at 37 °C with appropriate antibiotics. *Yptb* strains were cultured in Yersinia-Luria-Bertani (YLB) broth (tryptone 1%, yeast extract 0.5%, NaCl 0.5%) at 26 °C with appropriate antibiotics. Markerless chromosomal gene deletion in *Yptb* and *E. coli* was performed as described[21,47]. To construct the knock-out plasmid for deletion of *cccR* in *Yptb*, the *sg2O* fragment, 307 bp upstream fragment and the 325 bp downstream fragment of *cccR* were amplified using the primer pairs Δ*cccR*-sg20-F-SpeI/Δ*cccR*-sg20-R, Δ*cccR*−1F/Δ*cccR*−1R and Δ*cccR*−2F/Δ*cccR*−2R-SalI, respectively. The sg20, upstream and downstream PCR fragments were ligated by overlap PCR, and the resulting PCR product was digested with SpeI/SalI and inserted into pTargetF to produce pTargetF-Δ*cccR*. The knock-out plasmid pTargetF-Δ*fhuA* was constructed similarly. For expression, complementation and bacterial two-hybrid experiments, primers *cccR*-F-BamHI/*cccR*-R-SalI were used to amplify the *cccR* gene and the PCR product was digested with BamHI/SalI and inserted into pET28a, pGEX6p-1, pKT100, pUT18Cm and pKT100, respectively. The plasmids pET28a-*cccR*$_{1-220}$, pET28a-*cccR*$_{201-380}$, pGEX6p-1-*cccR*$_{1-220}$, pGEX6p-1-*cccR*$_{1-220}$, pKT100-*fhuA*, pET28a-*fhuA*, pET28a-*ftsZ*, pGEX6p-1-*ftsZ*, pGEX6p-1-*cccR*$_{1-110}$, pGEX6p-1-*cccR*$_{101-280}$ and pGEX6p-1-*cccR*$_{201-380}$ were constructed similarly. To construct pME6032-*cccR*, primers *cccR*-F-SacI/*cccR*-R-BglII were used to amplify the *cccR* gene and the PCR product of *cccR* was inserted into similarly digested pME6032. For site-directed mutagenesis, *cccR*$^{H192A}$−1F-BamHI/*cccR*$^{H192A}$−1R, *cccR*$^{H192A}$−2F/*cccR*$^{H192A}$−2R-SalI were used to amplify the upstream and downstream PCR fragments, then the fragments were ligated by overlap PCR, digested with BamHI/SalI and inserted into pET28a and pGEX6p-1. To construct the *lacZ* fusion reporter pDM4-*P$_{cccR}$::lacZ*, primers *P$_{cccR}$*-F-SalI/*P$_{cccR}$*-R-XbaI were used to amplify the promoter fragment and the PCR product was digested with XbaI/SalI and inserted into pDM4-*lacZ* to generate pDM4-*P$_{cccR}$::lacZ*.

### Protein toxicity assays
To assess the toxicity of CccR, *E. coli* BL21(DE3) strains containing pET28a derivatives expressing CccR variants were grown to logarithmic phase under non-inducing conditions and serially diluted 10-fold onto LB agar plates containing 0.1 mM IPTG with the LB agar plates without IPTG as the control. Plates were incubated at 30 °C for 16 h before being photographed. To assess the toxicity of purified CccR proteins, stationary phase *E. coli* strains grown in LB medium were collected, washed and diluted 40-fold into LB medium and treated with purified CccR proteins (250 ng/ml) for 6 h. After treatment, the cultures were serially diluted and plated onto LB agar plates, and colonies were counted after 12 h growth at 37 °C. The percentage of survival rate was calculated by dividing the number of CFU of treated cells by the number of CFU of cells without toxin treatment[21]. For growth curve measurement, *E. coli* strains were inoculated into LB liquid medium containing purified CccR proteins (250 ng/ml) and the

growth was monitored by measuring $OD_{600}$ at 1 or 2 h intervals. All these assays were performed in triplicate at least three times.

## Bacterial competition assays

Interspecies bacterial competition assays were performed according to previously described methods[21,35,48]. Overnight grown donor and recipient cells were washed and adjusted to $OD_{600}$ of 1.0 with LB medium. Mixtures of donor and recipient cells were established at 1:1. The cocultures were either spotted onto a 0.22 µm nitrocellulose membrane (Nalgene) placed on YLB agar plates at 26 °C for 48 h (for contact-dependent competition), or inoculated into 2 ml YLB medium at 26 °C with shaking for 24 h or 48 h (for contact-independent competition in liquid medium). For contact-independent competition performed on a solid surface, 5 µl of the recipient strain was spotted on 0.22 µm nitrocellulose membrane on YLB agar plates. After the bacterial solution was dried, another 0.22 µm nitrocellulose membrane was put on it and 5 µl of the donor strain was spotted in the same place as the second membrane and incubated at 26 °C for 48 h. Suspensions were serially diluted and plated on selective media for the quantification of CFUs. The *Yptb* donor and *E. coli* recipient strains were labeled with pKT100 (Km$^R$) and pBBRMCS5 (Gm$^R$), respectively, to facilitate screening. Competitive indices for each experiment were determined by dividing the final donor-to-recipient ratio by the initial donor. For competition assays between *Yptb* and *E. coli* in the mouse gut[21,49], 6-week-old BALB/c female mice were orally gavaged with streptomycin (100 µl of 200 mg ml$^{-1}$ solution) on day 1. On day 2, $5 \times 10^8$ CFU of *E. coli* DH5α containing pBBRMCS5-GFP was gavaged, and on day 3, $5 \times 10^8$ CFU of *Yptb* strains were orally gavaged. On day 4 and day 5, mice were sacrificed and cecum and small intestine tissue were separated, serially diluted, and spread on plates for CFU enumeration. The *Yptb* strains were screened on YLB plates containing nalidixic acid and *E. coli* strains carrying pBBRMCS5-GFP (Gm$^R$) were screened on LB plates containing gentamicin.

## Mass spectrometry

Mass spectrometry were performed according to described methods. The *E. coli* BL21(DE3) carrying pET28a-*ftsZ* and pGEX6P-1-*cccR* or pGEX6P-1-*cccR$^{H192A}$* were induced, collected, and crushed to purify the FtsZ protein. Purified proteins were separated by SDS-PAGE and gel slices containing the protein detected were digested as described[50]. The resulting peptides were resuspended in a solvent of 0.1% formic acid (v/v) and subjected to EASY-nLC 1000 interfaced via a Nanospray Flex ion source to an Orbitrap Fusion Tribrid mass spectrometer (Thermo Fisher Scientific). Peptides were loaded into a C18 trap column (C18, 3 µm particles, 100 µm ID, 3 cm length, Dr. Maisch GmbH) and the separation was carried out in a capillary C18 column (C18, 1.9 µm particles, 150 µm ID, 15 cm length, Dr. Maisch GmbH) at a flow rate of 500 nl/min with a 60 min LC gradient composed of Solvent A (0.1% formic acid (v/v)) and Solvent B (acetonitrile, 0.1% formic acid (v/v)). The gradient was 5-10% B for 2 min, 10–25% B for 40 min, 25-35% B for 9 min, 35-90% B for 4 min, and finally 90% B for 6 min. The mass spectrometer was operated in a data-dependent acquisition mode, in which the precursor MS1 scan (*m/z* 350–1550) was acquired in the Orbitrap at a resolution setting of 120,000, followed by Orbitrap HCD-MS/MS and ITHCD-MS/MS of the 20 most abundant multiply charged precursors in the MS1 spectrum. MS2 spectra were acquired at a resolution of 30,000. MS/MS data was processed using Mascot search engine (v.2.5.1, 2014, http://www.matrixscience.com; Matrix Science Ltd., UK). For precursor ions, the mass error was set to 10 ppm, and for fragment ions, the mass error was set to 0.02 Da.

## Biotin-17-ATP AMPylation assay

Biotin-17-ATP AMPylation assays were performed according to described methods[51]. Assays were performed using purified 1.5 µg GST-CccR or GST-CccR$^{H192A}$ and 5 µg FtsZ or FtsZ$^{T8A}$ in a reaction mixture containing 50 mM MOPS, 50 mM KCl, 5 mM MgCl$_2$, 500 µM biotin-17-ATP and 1 mM GTP. Biotinylation reactions were incubated at 37 °C for 30 min and terminated by the addition of EDTA and SDS loading buffer with 1% β-mercaptoethanol and boiled. The reaction products were divided into two parts, one of which was stained with coomassie bright blue after SDS-PAGE, and the other was resolved by SDS-PAGE and transferred to nitrocellulose membranes. The Streptavidin HRP blot was performed according to the manufacturer's protocol (Chemiluminescent Biotin-labeled Nucleic Acid Detection Kit, Beyotime). Membranes were washed with PBS and biotin-labeled blots were developed using ECL and visualized by chemiluminescence.

## GTPase assay

To measure the GTPase activity of FtsZ, all proteins were purified in Tris buffer. The GTPase Activity Assay Kit (MAK113, Sigma) was used in 96-well plates at room temperature. FtsZ was preincubated with the indicated amounts of CccR or CccR$^{H192A}$ in 20 µl assay buffer (40 mM Tris, 80 mM NaCl, 8 mM MgAc$_2$, and 1 mM EDTA, pH 7.5), and the final volume was adjusted to 30 ml with ddH$_2$O. Similar reactions without any protein or only one of the proteins being tested were used as controls. To test the dose-dependent activity of CccR, 5 µg FtsZ was preincubated with increasing amounts of CccR. The reactions were initiated by adding 10 µl of 4 mM ATP and incubated for 30 min before terminating with 200 µl of malachite green. After 30 min at room temperature, the intensity of the signal was measured by determining the absorbance at 620 nm. The concentration of free phosphate in the reactions was calculated from a standard curve using the phosphate standard supplied in the kit.

## Regulation analyses

The *lacZ* fusion reporter plasmid pDM4-$P_{cccR}$::*lacZ* was transformed into *E. coli* S17-1λpir and mated with *Yptb* stains according to described procedures[52]. The chromosomal pDM4-$P_{cccR}$::*lacZ* fusion reporter strain was grown in YLB broth to $OD_{600}$ 1.5 at 26 °C, and the β-galactosidase activities were assayed with *o*-nitrophenyl-β-galactoside (ONPG) as the substrate[53]. Effects of CccR proteins delivered from *Yptb* donor strains on $P_{cccR}$ promoter activity in the Δ*cccR*($P_{cccR}$::*lacZ*) recipient reporter strains were tested using the Corning Transwell 12-well multiwell system with a 0.4-µm-pore-size polyester membrane (Corning, New York, USA). This filter system was chosen to allow the transfer of secreted proteins but not of bacteria. Exponentially growing bacterial cells were washed and adjusted to $OD_{600}$ of 0.2 with YLB medium. A total of 1.5 ml of the Δ*cccR*($P_{cccR}$::*lacZ*) reporter strain was added to the bottom chambers of the base plate, and the filter system was mounted onto the base plate. 0.5 ml of different *Yptb* donor strains were added to the upper chamber. The LacZ activity in the recipient reporter strain was detected after coincubation at 26 °C for 12 h or 24 h. Electrophoretic mobility shift assay (EMSA) and DNase I footprinting were performed as described[52]. The isothermal titration calorimetry (ITC) experiment was carried out on a Nano ITC standard volume isothermal calorimeter (TA Instruments, New Castle, DE) to quantify the interactions between CccR variants and the $P_{cccR}$ promoter probe[54].

## RNA-seq experiments

Effects of CccR proteins delivered from *Yptb* donor strains on gene expression in the Δ*cccR* recipient strains were tested using the Corning Transwell 6-well multiwell system with a 0.4-µm-pore-size polyester membrane (Corning, New York, USA). Exponentially growing *Yptb* WT and Δ*cccR* cells were washed and adjusted to $OD_{600}$ of 0.2 with YLB. A total of 2.6 ml of the Δ*cccR* cells were added to the bottom chambers of the base plate (three biological replicates), and 1.5 ml of the *Yptb* WT and Δ*cccR* cells were added to the upper chamber, respectively. After co-incubation at 26 °C for 6 h, Δ*cccR* cells in the bottom chambers were collected for RNA-seq transcriptomics analysis. Total RNA was

extracted for cDNA library construction by using TRIzol Reagent/ RNeasy Mini Kit (Qiagen) and analyzed with the Bioanalyzer 2100 system (Agilent Technologies). 1 μg total RNA was used for library construction and libraries with different indices were multiplexed and loaded on an Illumina HiSeq/Novaseq instrument according to the manufacturer's instructions (Illumina, CA, USA). Sequencing was carried out using a $2 \times 150$ paired-end (PE) configuration. The result of sequencing was aligned with the reference genome of *Yptb* and RPKM (Reads per kilobase transcriptome per million mapped reads) was used to normalize the expression level of genes. The differential expressed genes were shown as fold change calculated by $log_2$ (RPKM of WT/ $\Delta cccR$). The Kyoto Encyclopedia of Genes and Genomes (KEGG) pathway analysis was used to investigate the role of differentially expressed genes, a pathway was considered significantly enriched when the *P* value was less than 0.05. Gene set enrichment analysis (GSEA) was carried out with hallmark gene sets.

## Microscopy

Overnight grown *E. coli* BL21(DE3) cells harboring pET28a, pET28a-*cccR* and pET28a-*cccR*[H192A] were diluted 100-fold into LB broth. The expression of recombinant proteins was induced by the addition of 0.5 mM IPTG at $OD_{600}$ 0.5. 10 μl samples at different time points were taken and dropped to a slide covered with 1% agarose in advance, and covered with the cover glass[55]. Samples were inspected in $63 \times$ oil objective with an inverted fluorescence microscope (Leica, DMi8, Germany) or in $100 \times$ oil objective with the high-speed rotary disc type fluorescence confocal microscope (Andor Revolution-XD, UK). To analyze FtsZ localization, overnight cultures of *E. coli* FtsZ-mVenus expressing pBAD22a-*cccR* were diluted at 1:100 and grown in liquid LB medium to an $OD_{600}$ of 0.5. Cells were harvested by centrifugation and washed twice with LB, then 0.2% L-arabinose was added. Cells were imaged using high-speed rotary disc type fluorescence confocal microscope. To observe CccR cell entry, Alexa Fluor 488-conjugated CccR was prepared as described[21,56]. 600 μl cultures at $OD_{600} = 1.2$ were washed three times, resuspended in M9 glucose containing 1 μM fluorophore-conjugated CccR, and incubated in the dark at room temperature for 30–40 min. The cells were washed five times to remove the free label and resuspended in 100 μl volume in M9 glucose. 10 μl of the cell suspension was dispensed onto 1% (w/v) agarose on a microscope slide before sealing with a clean cover glass. The result was obtained by high-speed rotary disc type fluorescence confocal microscope.

## FtsZ polymerization assay

*E. coli* FtsZ protein was diluted at a final concentration of 10 μM in MOPS-KOH (50 mM MOPS, 50 mM KCl, 5 mM MgCl$_2$, pH 6.5). The CccR protein was added to the protein at a series of concentrations (0, 0.1, 1, 10, 100 μM) and CccR[H192A] was used as a control. The polymerization reaction was initiated by adding 1 mM GTP and the mixtures were incubated at 25 °C for 1 h. Then the samples were centrifuged at $18,800 \times g$ for 60 min, and pellets were re-suspended in MOPS-KOH and analyzed with SDS-PAGE. Gels were stained with coomassie bright blue and the protein content of binding bands was measured by densitometric quantification using Image J software (NIH, Frederick, USA)[57].

## Negative stain electron microscopy

To visualize the effect of CccR on FtsZ polymerization, purified FtsZ protein (12.5 mM) was incubated in the assembly buffer (50 mM HEPES-KOH, pH 7.5, 300 mM KAc, 5 mM MgAc$_2$, 1 mM ATP) in the presence or absence of CccR (10 nM) at 30 °C for 30 min, with CccR[H192A] and DMSO as controls. FtsZ polymerization was initiated by the addition of 1 mM GTP and incubated for an additional 30 min. To visualize the effect of CccR on depolymerization of FtsZ filaments, preformed FtsZ filaments were coincubated with CccR protein (10 nM)

in the presence of 1 mM ATP at 30 °C for 30 min. The solution containing polymerized FtsZ was diluted 3 times with the assembly buffer. Diluted polymeric mixtures were adsorbed onto glow-discharged carbon-coated copper mesh grids for 60 s, stained with 2% phosphotungstic acid for 30 s, then washed once with assembly buffer and three times with deionized water, and allowed to air dry. Grids were imaged using the FEI Tecnai Spirit 120 kV electron microscope equipped with a 30,000 times Camera (FEI, USA). Polymer lengths in blinded frames from each sample were measured manually using the microscope scale bar.

## Crystallization

Crystallization was performed according to described methods[58,59]. The purity of CccR was ~95% as assessed by SDS-PAGE and initial crystallization screens of native CccR were conducted via sitting-drop vapor diffusion using commercial crystallization screens. The protein concentration used for crystallization was 5–7 mg/ml. Hampton Research kits were used in the sitting drop vapor diffusion method to get preliminary crystallization conditions at 16 °C. Crystallization drops contained 0.5 μl of the protein solution mixed with 0.5 μl of reservoir solution. To solve the phase problem, Se-Met was incorporated into CccR and the CccR[Se-Met] was purified similarly to native CccR except with the inclusion of 5 mM DTT added to the buffer during the purification process. The protein concentration of CccR[Se-Met] used for crystallization was also ~7 mg/ml. Diffraction-quality crystals of CccR were grown and optimized in the same condition. All crystals were flash-frozen in liquid nitrogen, with the addition of 20%–25% (v/v) glycerol as cryoprotectant.

## Data collection and structure determination

X-ray diffraction for Se-Met CccR and native CccR were collected at the beamline BL-17U1 of the Shanghai Synchrotron Radiation Facility (SSRF). All data were indexed and scaled using HKL2000 software[60]. The initial phase of CccR was determined by using the single-wavelength anomalous dispersion (SAD) phasing method. Phases were calculated using AutoSol implemented in PHENIX. AutoBuild in PHENIX was used to automatically build the atom model. Molecular Replacement was then performed with this model as a template to determine the structure of other complexes. After several rounds of positional and B-factor refinement using Phenix. Refine with TLS parameters alternated with manual model revision using Coot, the quality of final models was checked using the PROCHECK program. The quality of the final model was validated with MolProbity. Structures were analyzed with PDBePISA (Protein Interfaces, Surfaces, and Assemblies), Dali, and Details of the data collection and refinement statistics are given in Supplementary Table 1. All of the figures showing structures were prepared with PyMOL.

## Secretion assays

Secretion assays for CccR were performed according to described methods[61,62]. All samples used for secretion assays in this study were taken at mid-exponential phase corresponding to an $OD_{600}$ of 0.8-1.0. Briefly, strains were inoculated into 300 ml YLB broth and incubated with continuous shaking until $OD_{600}$ reached 0.85-0.9 at 26 °C. 2 ml culture was centrifuged and the cell pellet was resuspended in 100 μl SDS-sample buffer; this whole cell lysate sample was defined as Total. 290 ml of the culture was centrifuged, then the supernatant was filtered through a 0.22 μm filter (Millipore, MA, USA), and the proteins were extracted by filtration over a nitrocellulose filter (BA85) (Whatman, Germany) three times. The filter was soaked in 100 μl SDS sample buffer for 15 min at 65 °C to recover the proteins present, and the sample was defined as Secreted. All samples were normalized to the $OD_{600}$ of the culture and volume used in preparation. All the samples were separated in SDS-PAGE and the signals were observed by the western blot.

## Western blot analysis

Western blots were performed according to described methods[63,64]. Protein samples resolved by SDS-PAGE were transferred onto polyvinylidene fluoride membranes (Millipore). After blocking with QuickBlock™ Blocking Buffer (Shanghai Beyotime Biotechnology, China) for 8 h at 4 °C, membranes were incubated overnight at 4 °C with the appropriate primary antibody: rabbit anti-CccR (Laboratory preparation), 1:1000; rabbit anti-GST (Santa Cruz, cat# 53909), 1:1000; rabbit anti-RNAP (Santa Cruz, cat# sc-56766), 1:5000; mouse anti-His (Santa Cruz, cat# sc-8036), 1:5000. The membranes were then washed five times with TBST buffer (10 mM Tris-HCl, 150 mM NaCl, 0.05% Tween, pH 7.5) and incubated with 1:10,000 dilution of goat anti-rabbit horseradish peroxidase-conjugated secondary antibodies (DIYIBIO, China, cat# DY60202) or goat anti-mouse horseradish peroxidase-conjugated secondary antibodies (DIYIBIO, China, cat# DY60203) at 4 °C for 4 h. After another seven washes with TBST buffer, chemiluminescent signals were detected by using the ECL Plus Kit (GE Healthcare). Uncropped images of blots can be found in the Source Data file.

## Bacterial two-hybrid

Bacterial two-hybrid complementation assays were performed as described[65]. Briefly, the pKT25 and pUT18C derivatives were co-transformed into *E. coli* BTH101 and cultured on MacConkey plate (Ampicillin 100 μg·ml⁻¹, Kanamycin 50 μg·ml⁻¹, IPTG 1 mM) at 30 °C. At the same time, the plasmid pKT25-zip/pUT18C-zip and pKT25/pUT18C were co-transformed into *E. coli* BTH101 to serve as positive and negative controls, respectively. Interactions were tested using MacConkey medium and a red colony color shows an interaction between proteins, while a white colony color attests the absence of interaction. Efficiency of interactions between different proteins were quantified by measuring β-galactosidase activities in liquid cultures.

## Data quantitation and statistical analyses

Statistical tests, the number of events quantified, the standard deviation of the mean, and statistical significance are reported in figure legends. Statistical analysis was performed using GraphPad Prism9 software (GraphPad Software, San Diego California USA) and statistical significance is determined by the value of $p < 0.05$.

## Reporting summary

Further information on research design is available in the Nature Portfolio Reporting Summary linked to this article.

# Data availability

The atomic coordinates and structure factors of the CccR have been deposited in the Protein Data Bank (PDB) under accession code 7XUX. RNA-seq raw FASTQ files for the RNA-seq libraries have been deposited in the NCBI Sequence Read Archive (SRA) under the code BioProject accession PRJNA905726. All the other data that support the findings of this study are available within the paper and its Supplementary Information and Supplementary Data, or from the corresponding authors upon request. Source data are provided with this paper.

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

## Acknowledgements

This work was supported by the grant of National Key R&D Program of China (2018YFA0901200 to X.S., 2021YFC2301403 to S.O.), the National Natural Science Foundation of China (31725003 to X.S., 82225028 and 82172287 to S.O., 31970114 and 32170130 to Y.W., 32000022 to H.G., and 32100149 to L.Z.), the Special Funds of the Central Government Guiding Local Science and Technology Development (2020L3008 to S.O.), and the High-level personnel introduction grant of Fujian Normal University (Z0210509 to S.O.). We thank Dr. Sheng Yang at Institute of Plant Physiology and Ecology, Chinese Academy of Sciences for providing the CRISPR-Cas9 System, and Dr. Harold P. Erickson at Duke University Medical Center and Dr. Yaodong Chen at Northwestern University for providing *E. coli* strain FtsZ-FP (*ftsZ* – 55-mVenus-56). We also thank Dr. Jingfang Liu and Weilin Li (Public Technology Service Center Institute of Microbiology, Chinese Academy of Sciences) for help in identification of AMPylation sites with mass spectrometry, and the Teaching and Research Core Facility at College of Life Science and Life Science Research Core Services, NWAFU for technical support.

## Author contributions

X.S., D.W. and L.Z. conceived the project. D.W., L.Z., D.Y., C.L., Y.C., Y.Q., H.G., and L.X. performed experimental work; X.Z., H.W and S.O. determined the structures and analyzed protein properties using biophysical tools; X.L, Y.Y, C.W. and Y.W. performed the computational analyses and

provided technical support; D.W., L.Z., S.O. and X.S. wrote the manuscript. X.S. supervised the study. All authors discussed the results and commented on the manuscript.

## Competing interests

The authors declare no competing interests.
