## [Peer Review File · Nature Communications]

A secreted effector with a dual role as a toxin and as a transcriptional factorReviewer #1 (Remarks to the Author):

In the study "Cell-to-cell communication mediated by a T6SS secreted transcriptional regulator" , the authors use a multidisciplinary approach, including genetic, biochemical and biochemical assays, to functionally and structurally characterize the CccR toxin of *Yersinia pseudotuberculosis* (Yptb). CccR is a FIC protein that additionally presents an HTH domain, so it can act as a transcription factor. In the manuscript the authors show that the protein is actually bifunctional, since once excreted into the medium through the T6SS system, it can act as a toxin in competing bacteria by the AMPylation of the cell division protein FtsZ. To mediate this function CccR is imported in the target bacteria engaged to FhuA. Meanwhile, CccR functions as a signaling protein in kin bacteria acting as a transcriptional factor. The work is multidisciplinary and very well executed, the results are clear, the manuscript is well written and the logic is easy to follow.

Taken together, the novelty and originality of the results presented as well the large number and high technical quality of experimental approaches used to confirm these results, the manuscript shows the sufficient entity to be published in Nature Communications. However, in my view, some of the (main) conclusions of the manuscripts are not completely supported and/or checked and, therefore, the manuscript needs additional data addressing the following concerns before to be accepted for the publication.

Once of the major contributions of the study is to demonstrate the bifunctional character of CccR, acting as both a toxin and a transcriptional regulator in receptor cells. With these results, the authors propose a model for the dual role of CccR in bacterial competition and cell-to-cell communication (fig 6) where the toxin activity is working in nonself recipient cells (competitors) meanwhile the signaling activity (transcriptional regulator) acts in kin cells. However, the selectivity of this dual activity is supported by the fact that the authors have not tested both functions in both types of cells (competitor and kin). In other words, could CccR act as a toxin in kin cells? or, oppositely, could CccR act as a transcriptional regulator in competitor cells?

The authors have basically all the tools to confirm both points and thus support the proposed model.

1. Toxin activity in Yptb, it is easy to evaluate whether overexpression of CccR induces filamentation of this bacterium or whether CccR mediates in vitro the AMPylation of Yptb FstZ. In addition, the authors know the AMPylation site for *E. coli* FstZ (Thr8), so they could evaluate in silico how conserved is this site in other strains and then discuss about possible range of target strains.

2. Regulatory (signaling) activity in competitor cells. The authors make a great effort throughout the manuscript to map the DNA binding site of CccR, as well as to unveil the genes regulated by this protein in Yptb. However, with all this information available they do not evaluate whether CccR can regulate genes in competing strains once it is imported. If the authors know the DNA binding site, they could locate these sequences in the *E. coli* genome and design assays with reporters, such as those used in the study, to evaluate whether regulation also exists for *E. coli* genes.

In addition, and related to the regulatory capacity of CccR, the authors do not make an intensive analysis of the RNAseq data. Basically, it is a description of the experimental results indicating the number of up- and down-regulated genes and the pathways in which these genes are involved. It is quite possible that a more detailed analysis will be presented in future papers, but the authors should discuss the data in more depth here. At least the authors should localize the CccR binding boxes in the regulated genes to assess whether it is a direct or indirect regulation, even more so when CccR seems to act as a transcriptional repressor and the RNAseq results show a large number of up-regulated genes by CccR.

Also related with the regulatory activity of CccR, the authors show from the structural

data the weak dimerization surface of the protein, which is completely abolished with mutations at only two positions (N13 and K175). The authors could analyze whether this mutant is defective in regulatory capacity (CccR should be dimer to recognize the palindromic binding site), but not in toxin activity. This data would reinforce the independence of both activities and may imply additional levels of regulation for the system.

Finally, and related with the previous point, I consider that the authors make little use of the structural data (which must have required a major effort), other than to explain structurally the infrequent DNA-binding site (two inverted repeats of the 10-bp separated by a spacer of 26 bp) of CccR. The authors could also structurally confirm the catalytic activity of this protein by docking the substances and justifying why the H192A mutation (widely used in the studies) is catalytically inactive.

Reviewer #2 (Remarks to the Author):

In this exciting study, the authors made an unusual discovery of the function of a T6SS in *Yersinia pseudotuberculosis*: it secretes the multifunctional protein CccR into the extracellular milieu and the protein is taken up by competitive or kin cells via binding to the transporter FhuA. Most interestingly, whereas CCCR functions as an AMPylator to inhibit cell division by attacking FtsZ, it acts as a transcriptional regulator to modulate the expression of a large set of genes, making the bacterium better fit in hostile environments. Overall, this is an excellent study, the experiments are well-designed, and the results are of high quality. The findings have significantly expanded our appreciation of the function of T6SSs and their substrates.

Specific comments:

1. FtsZ is highly conserved among different bacterial species. How does CccR distinguish FtsZ of *Yptb* from that of other bacteria such as *E. coli*? Can CccR AMPylate the *Yptb* FtsZ in biochemical reactions?

2. Another weakness is the Discussion. The authors should at least speculate the reason for a few important issues. For example, why delivered CccR is necessary for the recipient cells? Do they not produce endogenous CccR under certain conditions? If so, why do the donor cells do?

Reviewer #3 (Remarks to the Author):

I read with great interest the manuscript entitled "Cell-to-cell communication mediated by a T6SS secreted transcriptional regulator". The authors of this study demonstrate that CccR, a T6SS effector from *Yersinia pseudotuberculosis*, acts as a contact-dependent and contact-independent toxin against prey cells as well as exerts transcriptional control on sister cells in the producing population. Overall, this is a solid study presenting novel and exciting results that make a significant contribution to the T6SS field.

The majority of the paper is dedicated on the action of CccR on prey cells. The experiments performed in this part of the study are excellent and beautifully presented. The authors back each of their claims with multiple experiments using diverse approaches and complementary techniques that are well-designed and executed. For this part of the work, I only have a few minor suggestions for improvement:

- It would be useful to have some quantifications for the data presented in figures 1d, 2e and extended data figures 1a and 1f. It is clear that the cells are elongated and

filamented upon exposure to CccR, nonetheless a graph reporting on cell length would make it more compelling.

- Line 119: The phrase "These results demonstrated that CccR is a FIC toxin." is rather absolute. I would re-phrase to "These results show that CccR exerts FIC toxicity when expressed in *E. coli*".

- Why did the authors choose to use DH5 alpha cells in their colonization assays. This is a strange choice of strain for in vivo experiments so some justification would be useful.

- Line 183: The fact that the N-terminal portion of CccR is important for internalization into prey cells further support that this toxin behaves in a colicin-like manner when it comes to prey cell entry. This could be indicated in the text.

A smaller part of the manuscript is dedicated on the action of CccR on the producing population and its sisters and shows proof that this effector acts as a negative autoregulator. In my view, the data presented on the activity of the effector on kin cells is robust. That said, this section left me with many unanswered questions, some of which could be investigated by the authors, especially considering that this is the main finding displayed in the title of the manuscript. Please find my questions and suggestions below:

- It is not discussed whether CccR has an immunity protein. Indeed, it must not since it can act on the producer and its sisters. In this case, how does the producer counteract the action of this toxin on its own FtsZ? Does CccR not interact with FtsZ from *Yersinia pseudotuberculosis*? Or does CccR not have any activity against that FtsZ analogue? The authors are experts in all of the techniques required to investigate whether CccR acts on *Yersinia pseudotuberculosis* FtsZ. In addition, in the case where this toxin does not act on the target of its producer, I think it would strengthen the manuscript to show that it does act of FtsZ analogues of other prey bacteria beyond *E. coli*.

- When CccR acts on the producer does that happen while it is produced inside the cell, or does it get re-internalized after secretion through the T6SS? It is somewhat shown that it is the latter using the reporter strain. That said, identifying the receptor in *Yersinia* and doing similar experiments as the Transwell assays presented currently in the manuscript with a mutant strain for that receptor would give more definitive answers, since both scenarios could be at play. Once more, the authors have the expertise to investigate this.

- I can see that this toxin affects the producer in a very specific manner and the authors discuss this in the context of fitness of the producer cell (Fig. 6). However, the autoregulatory function of CccR in the absence of an immunity protein could also be protective of the producer and its sisters. The authors do not touch on what part of the T6SS apparatus delivers this toxin. But one can imagine a scenario where while it is produced, CccR binds to a T6SS component (Hcp or VgrG) and that prevents it from being toxic. So, its repressor function could just aim to prevent toxicity in cases where its binding to T6SS components is not sufficient for some reason. The same can be said for sister cells; in the presence of many prey cells a lot of this toxin will be internalized to kill them. In the absence of prey, it could be acting as a signaling molecule to inform the population that its production is not necessary since it is mainly sisters that internalize it. Those hypotheses do not negate that it is acting as a signaling molecule in addition to being a toxin. I just feel that the discussion is particularly brief and is using a very specific angle when it comes to cell-to-cell communication; it should be re-worded to be more inclusive of other possibilities that could still belong to the realm of cell-to-cell communication. Finally, if the authors wish to claim that the repressor function of CccR increases fitness of the producer they could try and show this using their colonization model and producers with wild-type regulation, as well as disrupted regulation (for example using their palindrome sequences) and assess how well these

two strains perform against *E. coli* in vivo.

Overall, I would like to re-iterate that this is a very nice study and with some changes it could made a big contribution to the T6SS field. Thank you for the opportunity to review this very nice paper!

Reviewer #4 (Remarks to the Author):

This manuscript focuses on a *Yersinia pseudotuberculosis* T6SS secreted effector, CccR, containing two domains: a FIC domain and a HTH domain. In a systematic manner the authors demonstrated that this effector functions as a FIC toxin which mediates growth inhibition, it is secreted via T6SS-3 and 4 in a contact-dependent and contact-independent manner. In relation to its contact-independent effect, it was found that FhuA mediates CccR entry into target cells. The main target of CccR in *E. coli* was found to be FtsZ. CccR was found to bind and AMPylate FtsZ, thus leading to inhibition of the FtsZ GTPase activity and its polymerization. Additionally, the crystal structure of CccR was determined, it was found to autoregulate its own expression using an inverted-repeat sequence located in its regulatory region. Moreover, CccR was shown to regulate gene expression also after it was delivered into neighboring kin cells. Thus, the authors demonstrated that CccR functions on competing cells (growth inhibition using its FIC domain) as well as on kin cells (transcriptional regulation using its HTH domain).

Major comments:

1. Fig. 3D and Fig. 3E, in both panels the effect of CccR on FtsZ was examined. In Fig. 3D the CccR effect on FtsZ GTPase activity and in Fig. 3E the CccR effect on FtsZ polymerization. In both assays, the concentration of CccR used was very high, and this does not fit the function of CccR as a protein harboring enzymatic activity – AMPylation. In the GTPase activity assay in order to reduce the FtsZ activity a ratio of 1:1 between the substrate (FtsZ) and enzyme (CccR) was required. The requirement for such a ratio is expected in the case where the binding of CccR to FtsZ per se results in the reduction of FtsZ GTPase activity but not if CccR enzymatic activity makes the effect. In the FtsZ polymerization assay, there was a need for 100µM of CccR in order to inhibit the polymerization of 10µM of FtsZ, this clearly does not fit the hypothesis that AMPylation of FtsZ results in the inhibition of polymerization, but it rather seems that the binding between the two proteins led to the observed inhibition. This issue should be resolved for example by using low concentration of CccR over a time course to examine its effect on FtsZ, or to use clean AMPylated-FtsZ in the two assays (without CccR).

2. Fig. 5C (lines 339-344) – The authors generated an *E. coli* strain with the cccR gene containing a mutation in the palindromes of the regulatory element (PcccRm) which drastically increased CccR expression. The protein band in Fig. 5C in the relevant lane is very strong. How such a strain can be alive? The strain produces high level of a potent toxin which AMPylates FtsZ, how can this strain replicate? this result does come together with the other information presented in the manuscript regarding CccR effect on *E. coli*.

3. The authors identified the regulatory element recognized by CccR, which is rather special (line 329). The RNA-seq analysis resulted in the identification of 447 genes which were up- or down regulated after CccR delivery. It is highly likely that very few (if any) of these genes harbor the CccR regulatory element, therefore it is not clear how CccR affects their expression. It might be that CccR function as a FIC toxin is responsible for the effect (induction of stress) and not its function as a transcriptional regulator. The RNA-seq analysis should be repeated with a CccR mutated in the FIC domain to validate that the change in gene expression occurs due to CccR function as a transcriptional regulator.

4. It was not described in the manuscript if there is a known antitoxin for CccR. The FtsZ

proteins of *E. coli* and *Yersinia pseudotuberculosis* are identical at the N-termini where CccR AMPylates FtsZ. Therefore, it is expected that *Yersinia pseudotuberculosis* will harbor an anti-toxin which inhibit CccR activity. Is there any information about the CccR antitoxin?

Response to Reviewers

Reviewer #1 (Remarks to the Author):

In the study "Cell-to-cell communication mediated by a T6SS secreted transcriptional regulator", the authors use a multidisciplinary approach, including genetic, biochemical and biochemistry assays, to functionally and structurally characterize the CccR toxin of *Yersinia pseudotuberculosis* (*Yptb*). CccR is a FIC protein that additionally presents an HTH domain, so it can act as a transcription factor. In the manuscript the authors show that the protein is actually bifunctional, since once excreted into the medium through the T6SS system, it can act as a toxin in competing bacteria by the AMPylation of the cell division protein FtsZ. To mediate this function CccR is imported in the target bacteria engaged to FhuA. Meanwhile, CccR functions as a signaling protein in kin bacteria acting as a transcriptional factor. The work is multidisciplinary and very well executed, the results are clear, the manuscript is well written and the logic is easy to follow.

Taken together, the novelty and originality of the results presented as well the large number and high technical quality of experimental approaches used to confirm these results, the manuscript shows the sufficient entity to be published in Nature Communications. However, in my view, some of the (main) conclusions of the manuscripts are not completely supported and/or checked and, therefore, the manuscript needs additional data addressing the following concerns before to be accepted for the publication.

Once of the major contributions of the study is to demonstrate the bifunctional character of CccR, acting as both a toxin and a transcriptional regulator in receptor cells. With these results, the authors propose a model for the dual role of CccR in bacterial competition and cell-to-cell communication (fig 6) where the toxin activity is working in nonself recipient cells (competitors) meanwhile the signaling activity (transcriptional regulator) acts in kin cells. However, the selectivity of this dual activity is supported by the fact that the authors have not tested both functions in both types of cells (competitor and kin). In other words, could CccR act as a toxin in kin cells? or, oppositely, could CccR act as a transcriptional regulator in competitor cells? The authors have basically all the tools to confirm both points and thus support the proposed model.

Response: Thank you for your positive feedback and insightful comments regarding our manuscript. As required, we have provided additional experimental data in the revised manuscript to address the concerns you raised.

1. Toxin activity in *Yptb*, it is easy to evaluate whether overexpression of CccR induces filamentation of this bacterium or whether CccR mediates in vitro the AMPylation of *Yptb*

FstZ. In addition, the authors know the AMPylation site for *E. coli* FstZ (Thr8), so they could evaluate *in silico* how conserved is this site in other strains and then discuss about possible range of target strains.

Response: As suggested, we have tested the toxin activity of CccR in kin cells (*Y. pseudotuberculosis*) and the results showed that expressing of CccR did not result in growth arrest and cell filamentation of this bacterium (**Respond Fig. 1a-b, new Supplementary Fig. 2a-b**), implicating the existence of an unknown immunity mechanism to protect the CccR producing kin cell from intoxication.

Respond Fig. 1. a-b, Effects of cccR expression on *Yptb* growth. Growth curve (**a**) and microscopic analysis (**b**) of *Yptb* cells overexpressing cccR with the pME6032 vector. **c**, Sequence alignment of FtsZ proteins from different intestinal bacterial strains. Thr⁸ is relatively conserve in bacteria of Enterobacteriaceae but not in other families. **d**, Representative blot using avidin-HRP to detect biotinylated proteins following incubation of FtsZ proteins from indicated strains with bio-17-ATP and CccR proteins. Coomassie bright blue staining is shown as a loading control.

Sequence alignment analysis showed that the Thr⁸ residue is conserved in bacteria of *Enterobacteriaceae* but not in other families (**Respond Fig. 1c, new Supplementary Fig. 6d**). Consistently, *in vitro* AMPylation assays showed that CccR can also modify FtsZs from *Yptb* and *Salmonella enterica* (**Respond Fig. 1d, new Supplementary Fig. 6e**).

We have added these results to the main text and discussed about possible range of target strains in the discussion section as follow: “It will be interesting to determine whether this FtsZ-targeting mechanism is conserved among *Enterobacteriaceae* species that contain the conserved Thr⁸ residue (**Lines 252-254**)”.

2. Regulatory (signaling) activity in competitor cells. The authors make a great effort

throughout the manuscript to map the DNA binding site of CccR, as well as to unveil the genes regulated by this protein in *Yptb*. However, with all this information available they do not evaluate whether CccR can regulate genes in competing strains once it is imported. If the authors know the DNA binding site, they could locate these sequences in the *E. coli* genome and design assays with reporters, such as those used in the study, to evaluate whether regulation also exists for *E. coli* genes.

Response: Thank you very much for your professional comments. In this study we demonstrated that CccR acts as a transcriptional regulator to auto-repress its own expression not only in its producing cells but also in neighboring kin cells via specific DNA binding, which support the notion that CccR plays a role in cell-to-cell communication.

However, although we have identified 447 genes which were up- or down regulated more than 1.2-fold after CccR delivery by transcriptomics analysis, and the transcriptomics results have been validated by qRT-PCR analysis, we cannot identify putative CccR binding elements in the promoter regions of these regulated genes. So, it is not clear how CccR affects their expression. It might either be that CccR un-specifically recognize these promoters as a transcriptional regulator, or be that CccR function as a FIC toxin is responsible for the effect (induction of stress) for part of these genes. It seems impossible to reveal the global regulatory mechanism of CccR on these genes in this manuscript, but we will work on this topic continually in the future.

Similarly, we cannot identify putative CccR binding elements from *E. coli* genome. Even we performed a transcriptomics analysis of delivered CccR on *E. coli* cells, we didn't get valuable information. Given that delivered CccR will inhibit growth of the prey cells, and there is no CccR binding elements can be identified in the *E. coli* genome, the transcriptional regulator activity of CccR in the prey cells might can be ignored.

We have discussed these issues in the discussion section as follow: "Previously, it was reported that a contact-dependent growth inhibition (CDI) system in *Burkholderia thailandensis* delivers BcpA, a DNase toxin, to induce changes in the differential expression of 841 genes in immune target cells. Although the molecular mechanism underlying this process, termed contact-dependent signaling (CDS), requires the catalytic activity of BcpA, the details remain unclear. In this study, we provide direct evidence of a role of CccR as a transcriptional regulator that regulates gene expression by directly binding DNA. RNA-seq analysis resulted in the identification of 447 genes that were up- or downregulated more than 1.2-fold after CccR delivery. However, because we cannot identify putative CccR binding elements in the promoter regions of these regulated genes, it is not clear how CccR globally affects their expression. CccR may nonspecifically recognize promoters as a transcriptional regulator, or may function as an FIC toxin that is responsible for the effect (induction of stress). It will be interesting to determine whether CccR can act as a transcriptional regulator in prey cells. However, given that toxic CccR will inhibit prey cell growth, and no CccR

binding elements have been identified in the *E. coli* genome, transcriptional regulator activity of CccR in prey cells might can be ignored like BcpA (Lines 477-493)."

3. In addition, and related to the regulatory capacity of CccR, the authors do not make an intensive analysis of the RNAseq data. Basically, it is a description of the experimental results indicating the number of up- and down-regulated genes and the pathways in which these genes are involved. It is quite possible that a more detailed analysis will be presented in future papers, but the authors should discuss the data in more depth here. At least the authors should localize the CccR binding boxes in the regulated genes to assess whether it is a direct or indirect regulation, even more so when CccR seems to act as a transcriptional repressor and the RNAseq results show a large number of up-regulated genes by CccR.

Response: We thank the reviewer for this very insightful point. Although we have convincingly demonstrated that CccR acts as a transcriptional regulator to auto-repress its own expression not only in its producing cells but also in neighboring cells via direct DNA binding, we totally agree with you that the analysis of the transcriptomics results is not in-depth. The main problem is that we cannot identify putative CccR binding elements in the promoter regions of these regulated genes. So, it is not clear how CccR affects their expression. It might either be that CccR un-specifically recognize these promoters as a transcriptional regulator, or be that CccR function as a FIC toxin is responsible for the effect (induction of stress) for part of these genes.

It seems impossible to reveal the global regulatory mechanism of CccR on these genes in this manuscript, but we will work on this topic continually in the future. Possibly we need to perform a ChIP-seq to identify genes direct regulated by CccR. So, in this manuscript we mainly focused on the specifically regulation of CccR on its own expression, and these results are sufficient to support the role of CccR in cell-to-cell communication. We toned down the globally regulation part of CccR in the main text and the updated model (**Respond Fig. 2, new Fig 6**). As suggested, we have also discussed the CccR mediated regulation in-depth in the discussion section (**Lines 477-493**).

Respond Fig. 2. A model for the dual role of CccR in bacterial competition and cell-to-cell communication.

In CccR-producing donor cells, CccR negatively autorepresses its own expression with a feedback mechanism. Under competition conditions, CccR is released into the extracellular milieu to relieve the autorepression. Released CccR enters target cells by engaging the TonB-dependent outer membrane transporter FhuA in a bacteriocin-like manner. In the presence of many prey cells (Left), secreted CccR is preferentially internalized by prey cells to AMPylate the cell division protein FtsZ, leading to cell filamentation and growth arrest. Following the decrease in prey cells (Right), secreted CccR is internalized by sister cells to repress *cccR* expression by acting as a transcriptional regulator, informing the population that its production is unnecessary. Under this condition, CccR may also act as a global regulator that regulates expression of genes involved in iron acquisition, motility, and energy production to coordinate bacterial behaviors and increase bacterial fitness. Given that toxic CccR will inhibit the growth of prey cells, the transcriptional regulator activity of CccR in prey cells can be ignored.

4. Also related with the regulatory activity of CccR, the authors show from the structural data the weak dimerization surface of the protein, which is completely abolished with mutations at only two positions (N13 and K175). The authors could analyze whether this mutant is defective in regulatory capacity (CccR should be dimer to recognize the palindromic binding site), but not in toxin activity. This data would reinforce the independence of both activities and may imply additional levels of regulation for the system.

Response: Thank you for your professional comments. To analyze whether the CccR^{N13A/K175A} mutant is defective in regulatory capacity, we evaluated its effect on repressing the activity of the *P_{cccR}* promoter. As shown in Respond Fig. 3a (new Fig. 5a), while complementation of wild-type *cccR* gene in the $\Delta cccR$ mutant fully restored the repression effect on the *P_{cccR}* promoter, complementation of the *cccR*^{N13A/K175A} mutant gene failed to restore the repression effect. In addition, the binding affinity of the CccR^{N13A/K175A} mutant to the palindromic binding site dramatically reduced (Respond Fig. 3b, new

Supplementary Fig. 9a). These results are consistent with the prediction that CccR should be dimer to recognize the palindromic binding site. We also examined the toxicity of the CccR^{N13A/K175A} mutant by expressing it in *E. coli* and examined the modification of FtsZ by the CccR^{N13A/K175A} mutants. As shown in Response Fig. 3c-d (new Fig. 9b-c), the mutation has marginal effect on its toxicity and modification. These data reinforced the independence of both activities and have been included in the maintext (Lines 401-412).

Respond Fig. 3. Effect of dimerization plane mutation on CccR function.

a, β -Galactosidase activity analyses of *cccR* promoter activity using the transcriptional *PcccR::lacZ* chromosomal fusion reporter expressed in the *Yptb* WT, Δ *cccR* mutant, Δ *cccR(cccR)* and Δ *cccR(cccR^{N13A/K175A})* grown to stationary phase.

b, The ITC fitting results of CccR^{N13A/K175A} protein with *cccR* promoter probe. The thermodynamic data were collected from injections of *PcccR* into CccR^{N13A/K175A}, and the binding isotherm was fitted to a one-site binding model after subtraction of blank titration heats.

c, CccR^{N13A/K175A} is toxic to *E. coli*. Growth of *E. coli* BL21(DE3) cells containing a vector control or a vector expressing CccR or CccR^{N13A/K175A} under noninducing (no IPTG) or inducing (100 μM IPTG) conditions.

d, Representative blot using avidin-HRP to detect biotinylated proteins following incubation of *E. coli* FtsZ with bio-17-ATP and CccR or CccR^{N13A/K175A} proteins. Coomassie bright blue staining is shown as a loading control.

5. Finally, and related with the previous point, I consider that the authors make little use of the structural data (which must have required a major effort), other than to explain structurally the infrequent DNA-binding site (two inverted repeats of the 10-bp separated by a spacer of 26 bp) of CccR. The authors could also structurally confirm the catalytic activity of this protein by docking the substances and justifying why the H192A mutation (widely used in the studies) is catalytically inactive.

Response: Thank you for your professional and helpful comments. We have performed docking analysis and analyzed the structural data in depth as suggested as follow: "FIC proteins perform adenylation, which is dependent on the small-molecule nucleotide substrate ATP and transfer AMP groups; the FIC motif HXFX(D/E)GNRXXR is important

for recognition and catalysis of the substrate. Molecular docking showed that the active center of CccR bound ATP, and H₁₉₂ and R₂₀₃ in the FIC motif (H₁₉₂PFGNGNGR₂₀₃) formed hydrogen bonds with ATP (**Respond Fig. 4a, new Fig. 4d**). The conserved H₁₉₂ residue plays a key role in catalysis and its imidazole group deprotonates and activates the hydroxyl group for nucleophilic attack of the high-energy pyrophosphate bonds of the nucleotide substrate. The R₂₀₃ residue in the FIC motif could bind the ribose ring and γ -phosphate of nucleotides through the guanidine group, thus playing a role in the recognition of small-molecule nucleotide substrates. As a family of toxic proteins, most FIC proteins are expressed in an inhibited form autoinhibited by a conserved motif (S/T)XXE(G/N) termed the inhibitory α -helix (α_{inh}), which obstructs the active site to prevent optimal positioning of the ATP substrate for AMP transfer. Based on the location of the α_{inh} , FIC proteins were grouped into three classes: the α_{inh} located on an interacting protein reminiscent of toxin-antitoxin modules (Class I), or located N- terminal (Class II) or C- terminal (Class III) to the FIC motif within the same polypeptide chain. As a Class II FIC protein, CccR contains an N terminal α_{inh} (S₅₉ARIEG). The crystal structure of CccR shows S₅₉ and E₆₃ form a stable hydrogen bond with R₂₀₃ to prevent binding of the ATP substrate for modification (**Respond Fig. 4b, Supplementary Fig. 7e**) (Lines 332-351)".

Respond Fig. 4. a, Molecular docking of CccR with ATP. The conformation with the lowest docking energy was determined using Chimera software. Key residues of CccR involved in ATP binding are shown as purple sticks, and ATP is shown as green sticks. Two potential hydrogen bonds are indicated by dashed lines. **b,** The relationship between the inhibitory helix and the active center. The S59 and E63 residue of inhibitory helix (purple) form two potential hydrogen bonds with R203 (blue) in the active center to prevent binding of the ATP substrate for modification.

Reviewer #2 (Remarks to the Author):

In this exciting study, the authors made an unusual discovery of the function of a T6SS in *Yersinia pseudotuberculosis*: it secretes the multifunctional protein CccR into the extracellular

milieu and the protein is taken up by competitive or kin cells via binding to the transporter FhuA. Most interestingly, whereas CCCR functions as an AMPylator to inhibit cell division by attacking FtsZ, it acts as a transcriptional regulator to modulate the expression of a large set of genes, making the bacterium better fit in hostile environments. Overall, this is an excellent study, the experiments are well-designed, and the results are of high quality. The findings have significantly expanded our appreciation of the function of T6SSs and their substrates.

Response: We would like to thank the reviewer for the very positive comments on our study. We have revised our manuscript in accordance with the reviewer's comments.

Specific comments:

1. FtsZ is highly conserved among different bacterial species. How does CccR distinguish FtsZ of *Yptb* from that of other bacteria such as *E. coli*? Can CccR AMPylate the *Yptb* FtsZ in biochemical reactions?

Response: We thank the reviewer for this very insightful point. As you predicted, CccR can AMPylate *Yptb* FtsZ in biochemical reactions *in vitro* (please refer to Respond Fig. 1d, or new Supplementary Fig. 6e), suggesting that it cannot distinguish FtsZ of *Yptb* from that of other bacteria such as *E. coli*. However, the underlying immunity mechanisms that protect the CccR-producing *Yptb* cells from CccR toxicity remain to be revealed in the future, and we have discussed the possible immunity mechanisms in the discussion section (Lines 494-519).

2. Another weakness is the Discussion. The authors should at least speculate the reason for a few important issues. For example, why delivered CccR is necessary for the recipient cells? Do they not produce endogenous CccR under certain conditions? If so, why do the donor cells do?

Response: We thank the reviewer for these valuable suggestions which greatly improved our manuscript. We have re-written the discussion section and proposed an updated model (please refer to Respond Fig. 2, or new Fig. 6) which reasonably answered these important questions. We have proposed the new model as follows: "In CccR-producing donor cells, CccR negatively autorepresses its own expression with a feedback mechanism. Under competition conditions, CccR is released into the extracellular milieu to relieve the autorepression. Released CccR enters target cells by engaging the TonB-dependent outer membrane transporter FhuA in a bacteriocin-like manner. In the presence of many prey cells (Left), secreted CccR is preferentially internalized by prey cells to AMPylate the cell division protein FtsZ, leading to cell filamentation and growth arrest. Following the decrease in prey

cells (Right), secreted CccR is internalized by sister cells to repress *cccR* expression by acting as a transcriptional regulator, informing the population that its production is unnecessary. Under this condition, CccR may also act as a global regulator that regulates expression of genes involved in iron acquisition, motility, and energy production to coordinate bacterial behaviors and increase bacterial fitness. Given that toxic CccR will inhibit the growth of prey cells, the transcriptional regulator activity of CccR in prey cells can be ignored".

Reviewer #3 (Remarks to the Author):

I read with great interest the manuscript entitled "Cell-to-cell communication mediated by a T6SS secreted transcriptional regulator". The authors of this study demonstrate that CccR, a T6SS effector from *Yersinia pseudotuberculosis*, acts as a contact-dependent and contact-independent toxin against prey cells as well as exerts transcriptional control on sister cells in the producing population. Overall, this is a solid study presenting novel and exciting results that make a significant contribution to the T6SS field.

The majority of the paper is dedicated on the action of CccR on prey cells. The experiments performed in this part of the study are excellent and beautifully presented. The authors back each of their claims with multiple experiments using diverse approaches and complementary techniques that are well-designed and executed. For this part of the work, I only have a few minor suggestions for improvement:

Response: We would like to thank the reviewer for the very positive comments on our study. We have revised our manuscript in accordance with the reviewer's comments.

- It would be useful to have some quantifications for the data presented in figures 1d, 2e and extended data figures 1a and 1f. It is clear that the cells are elongated and filamented upon exposure to CccR, nonetheless a graph reporting on cell length would make it more compelling.

Response: The quantitation for Figures 1d and 2e was provided in new **Supplementary Fig. 1a and 5d**, respectively. The quantitation for extended data Figures 1a and 1f was provided in new **Supplementary Fig. 1a and 3d**, respectively.

- Line 119: The phrase "These results demonstrated that CccR is a FIC toxin." is rather absolute. I would re-phrase to "These results show that CccR exerts FIC toxicity when expressed in *E. coli*".

Response: Changed as suggested.

- Why did the authors choose to use DH5 alpha cells in their colonization assays. This is a strange choice of strain for in vivo experiments so some justification would be useful.

Response: Thank you very much for your professional comments. We agree with you that *E. coli* MG1655 was usually used for colonization assays. However, the MG1655 derived DH5 α have also been reported to be used in colonization assays (Song *et al.*, 2021; Sana *et al.*, 2016). In order to keep consistent with the *in vitro* competition results, we used DH5 α in our *in vivo* competition assays. The result seems reasonable and has not been affected by strain difference.

Relevant References:

1. Song L., *et al.* Contact-independent killing mediated by a T6SS effector with intrinsic cell-entry properties. *Nat Commun* (2021). 12 (1): 423.
2. Sana TG, *et al.* *Salmonella Typhimurium* utilizes a T6SS-mediated antibacterial weapon to establish in the host gut. *Proc Natl Acad Sci U S A.* (2016). 113(34):E5044-51.

- Line 183: The fact that the N-terminal portion of CccR is important for internalization into prey cells further support that this toxin behaves in a colicin-like manner when it comes to prey cell entry. This could be indicated in the text.

Response: Indicated in the text as suggested (Lines 201-203).

A smaller part of the manuscript is dedicated on the action of CccR on the producing population and its sisters and shows proof that this effector acts as a negative autoregulator. In my view, the data presented on the activity of the effector on kin cells is robust. That said, this section left me with many unanswered questions, some of which could be investigated by the authors, especially considering that this is the main finding displayed in the title of the manuscript. Please find my questions and suggestions below:

- It is not discussed whether CccR has an immunity protein. Indeed, it must not since it can act on the producer and its sisters. In this case, how does the producer counteract the action of this toxin on its own FtsZ? Does CccR not interact with FtsZ from *Yersinia pseudotuberculosis*? Or does CccR not have any activity against that FtsZ analogue? The authors are experts in all of the techniques required to investigate whether CccR acts on *Yersinia pseudotuberculosis* FtsZ. In addition, in the case where this toxin does not act on the target of its producer, I think it would strengthen the manuscript to show that it does act on FtsZ analogues of other prey bacteria beyond *E. coli*.

Response: We thank the reviewer for raising this very important issue. As you predicted, we found that CccR can not only interact with *Yptb* FtsZ (Respond Fig. 5a-b, new Supplementary Fig. 6a, c), but also AMPlate *Yptb* FtsZ (Respond Fig. 5c, new Supplementary Fig. 6e), suggesting that it cannot distinguish FtsZ of *Yptb* from that of other bacteria such as *E. coli*. This finding is consistent with the notion that antibacterial T6SS effectors usually target conserved and essential components of bacterial cells, including peptidoglycan, membrane phospholipids, nucleic acids, NAD⁺ and ATP. As a conserved cellular component, the cell division protein FtsZ has also been reported to be ADP-ribosylated by the antibacterial T6SS effector Tre1 (Ting *et al.*, 2018).

Respond Fig. 5. CccR interact with and AMPlate *Yptb* FtsZ.

a, Direct interaction between CccR and *Yptb* FtsZ verified with an *in vitro* GST pull-down assay.

b, Interaction between CccR and *Yptb* FtsZ confirmed using bacterial two-hybrid assay. Interactions between CccR and *Yptb* FtsZ was assessed using MacConkey maltose plates (upper) and the β-galactosidase assay (lower). Error bars indicate ± SD (n = 3 biological replicates), with two-sided, unpaired Student's *t*-test. ***, *P* < 0.001.

c, Representative blot using avidin-HRP to detect biotinylated proteins following incubation of FtsZ proteins from indicated strains with bio-17-ATP and GST-CccR. Coomassie bright blue staining is shown as a loading control.

However, the immunity mechanism of CccR seems very complex and the involved immunity protein needs to be investigated in the future. We have discussed this important issue from points including the inhibitory α_{inh} , intrinsic or extrinsic factors to modulate the expulsion of the inhibitory α_{inh} , auto-repress of expression, and possible interacting T6SS components in the discussion section as follow: “While the activity of Class I FIC can be directly regulated by removing the interacting antitoxin, Class II and III FIC proteins may require complex intrinsic or extrinsic factors to modulate expulsion of the inhibitory α_{inh} . For example, the toxicity of some Class III FIC proteins is regulated by oligomerization, autoadenylation, and metal ions. Although Class II FIC proteins account for 80% of total FIC proteins, nearly nothing is known about their biological functions and activity modulation.

Here, we found that the Class II FIC protein CccR auto-represses its own expression by acting as a transcriptional regulator, providing a new perspective for understanding the immune mechanism of toxic proteins. However, future work is needed to investigate the identity of the intrinsic and/or extrinsic factors that maintain the α_{inh} in an inhibitory state in self cells, and trigger expulsion of the α_{inh} to relieve autoinhibition in nonself cells. While each antibacterial T6SS effector is often co-expressed with a cognate immunity protein encoded in a bicistron to protect the producing cells and sister cells from intoxication, no immunity protein for CccR was identified in its neighboring ORFs. However, one can imagine that when CccR is produced, it may be bound by T6SS components (Hcp, PAAR or unknown chaperones) to prevent it from being toxic. Its repressor function could aim to prevent toxicity in cases where its binding to T6SS components is insufficient. The same is true of sister cells. In the presence of many prey cells, secreted CccR proteins will be preferentially internalized by prey cells to kill them. After the prey cells are killed, secreted CccR protein can serve as a signal molecule to inform the population that its production is unnecessary, because it is mainly internalized by sister cells in these conditions. This hypothesis is partially supported by the findings that CccR exhibited significant higher affinity to *E. coli* FhuA than to *Yptb* FhuA and enter *E. coli* cells more efficiently than *Yptb* cells (Lines 494-519).”

Relevant References:

Ting SY, *et al.*. Bifunctional Immunity Proteins Protect Bacteria against FtsZ-Targeting ADP-Ribosylating Toxins. **Cell**. 2018;175(5):1380-1392.e14.

- When CccR acts on the producer does that happen while it is produced inside the cell, or does it get re-internalized after secretion through the T6SS? It is somewhat shown that it is the latter using the reporter strain. That said, identifying the receptor in *Yersinia* and doing similar experiments as the Transwell assays presented currently in the manuscript with a mutant strain for that receptor would give more definitive answers, since both scenarios could be at play. Once more, the authors have the expertise to investigate this.

Response: We have performed the suggested experiments by coincubation of $\Delta cccR$, $\Delta cccR\Delta fhuA$ or $\Delta cccR\Delta fhuA(fhuA)$ reporter strains containing the chromosomal $P_{cccR}::lacZ$ reporter fusion, respectively, with *Yptb* WT in separate wells of the Transwell system. As shown in Respond Fig. 6 (new Supplementary Fig. 10), deletion of the CccR receptor gene, *fhuA*, in the $\Delta cccR(P_{cccR}::lacZ)$ recipient abolished the repression effect of *Yptb* WT coincubated in separate wells of the Transwell system, and the repression effect was fully

restored by complementation of *fhuA* in the $\Delta cccR\Delta fhuA(P_{cccR}::lacZ)$ recipient, further corroborating that CccR acts as an intercellular transcriptional regulator to mediate cell-to-cell communication once it has entered target recipient cells (**Lines 431-437**).

Thus, CccR can act as a transcriptional regulator not only in its producing cells, but also in neighboring kin cells once re-internalized after secretion through the T6SS, confirming the prediction that both scenarios could be at play.

Respond Fig. 6. FhuA is required for delivered CccR to exert transcriptional regulator activity in kin cells. Effects of CccR delivered from the *Yptb* WT donor on P_{cccR} promoter activity in the $\Delta cccR$, $\Delta cccR\Delta fhuA$ and $\Delta cccR\Delta fhuA(fhuA)$ recipient reporter strains were determined with β -galactosidase assays. Transwells were used to separate the *Yptb* WT donor and indicated recipient reporter strains, and the LacZ activity in the recipient reporter strains was detected after coinubation at 26°C for 12 h. *P* values were determined using a two-sided, unpaired Student's *t*-test, and differences were considered significant at *P* < 0.05. ***, *P* < 0.001.

- I can see that this toxin affects the producer in a very specific manner and the authors discuss this in the context of fitness of the producer cell (Fig. 6). However, the autoregulatory function of CccR in the absence of an immunity protein could also be protective of the producer and its sisters. The authors do not touch on what part of the T6SS apparatus delivers this toxin. But one can imagine a scenario where while it is produced, CccR binds to a T6SS component (Hcp or VgrG) and that prevents it from being toxic. So, its repressor function could just aim to prevent toxicity in cases where its binding to T6SS components is not sufficient for some reason. The same can be said for sister cells; in the presence of many prey cells a lot of this toxin will be internalized to kill them. In the absence of prey, it could be acting as a signaling molecule to inform the population that its production is not necessary since it is mainly sisters that internalize it. Those hypotheses do not negate that it is acting as a signaling molecule in addition to being a toxin. I just feel that the discussion is particularly brief and is using a very specific angle when it comes to cell-to-cell communication; it should be re-worded to be more inclusive of other possibilities that could still belong to the realm of cell-to-cell communication. Finally, if the authors wish to claim that the repressor function of CccR increases fitness of the producer they could try and show this

using their colonization model and producers with wild-type regulation, as well as disrupted regulation (for example using their palindrome sequences) and assess how well these two strains perform against *E. coli* in vivo.

Response: Thank you very much for these excellent suggestions which greatly improved our manuscript! We have provided some new results and re-worded the discussion section to be more inclusive of other possibilities as follow: “However, one can imagine that when CccR is produced, it may be bound by T6SS components (Hcp, PAAR or unknown chaperones) to prevent it from being toxic. Its repressor function could aim to prevent toxicity in cases where its binding to T6SS components is insufficient. The same is true of sister cells. In the presence of many prey cells, secreted CccR proteins will be preferentially internalized by prey cells to kill them. After the prey cells are killed, secreted CccR protein can serve as a signal molecule to inform the population that its production is unnecessary, because it is mainly internalized by sister cells in these conditions. This hypothesis is partially supported by the findings that CccR exhibited significant higher affinity to *E. coli* FhuA than to *Yptb* FhuA and enter *E. coli* cells more efficiently than *Yptb* cells (**Respond Fig. 7, new Supplementary Fig. 12) (Lines 509-519).**”

Respond Fig. 7. CccR exhibited higher affinity to *E. coli* FhuA than to *Yptb* FhuA and enter *E. coli* cells more efficient than *Yptb* cells.

a-b, Competitive internalization of CccR protein by *E. coli* and *Yptb* cells. Unlabeled *E. coli* and mCherry-labeled *Yptb* cells were mixed in a 1:1 ratio and incubated with AF488-conjugated CccR (Green) for 1 h at 30 °C. After 3 times wash with PBS, the internalization of CccR in different cells were observed

by confocal microscopy (a), and the percentages of *E. coli* and *Yptb* cells that exhibited green fluorescence (indicating CccR internalization) were quantified (b).

c, Comparison of the interaction of CccR with FhuA from *E. coli* and *Yptb* with bacterial two-hybrid assay. Interactions were assessed using MacConkey maltose plates (upper) and the β -galactosidase assay (lower). Error bars indicate \pm SD (n = 3 biological replicates), with two-sided, unpaired Student's *t*-test. ***, $P < 0.001$.

Despite repeated attempts, we were unable to construct a regulation disrupted mutant using palindrome sequences (possibly because the auto-repress mechanism is crucial for immunity), which precluded us to assess the repressor function of CccR in increasing the fitness of the producer. So, we toned down this part in the main text and in the updated model by using dotted lines (please refer to Respond Fig. 2, or new Fig. 6). We proposed the new model as follow: "In CccR-producing donor cells, CccR negatively autorepresses its own expression with a feedback mechanism. Under competition conditions, CccR is released into the extracellular milieu to relieve the autorepression. Released CccR enters target cells by engaging the TonB-dependent outer membrane transporter FhuA in a bacteriocin-like manner. In the presence of many prey cells (Left), secreted CccR is preferentially internalized by prey cells to AMPylate the cell division protein FtsZ, leading to cell filamentation and growth arrest. Following the decrease in prey cells (Right), secreted CccR is internalized by sister cells to repress *cccR* expression by acting as a transcriptional regulator, informing the population that its production is unnecessary. Under this condition, CccR may also act as a global regulator that regulates expression of genes involved in iron acquisition, motility, and energy production to coordinate bacterial behaviors and increase bacterial fitness. Given that toxic CccR will inhibit the growth of prey cells, the transcriptional regulator activity of CccR in prey cells can be ignored".

Overall, I would like to re-iterate that this is a very nice study and with some changes it could made a big contribution to the T6SS field. Thank you for the opportunity to review this very nice paper!

Response: Thank you very much for your insightful and professional comments which greatly improved our manuscript!

Reviewer #4 (Remarks to the Author):

This manuscript focuses on a *Yersinia pseudotuberculosis* T6SS secreted effector, CccR, containing two domains: a FIC domain and a HTH domain. In a systematic manner the authors demonstrated that this effector functions as a FIC toxin which mediates growth

inhibition, it is secreted via T6SS-3 and 4 in a contact-dependent and contact-independent manner. In relation to its contact-independent effect, it was found that FhuA mediates CccR entry into target cells. The main target of CccR in *E. coli* was found to be FtsZ. CccR was found to bind and AMPylate FtsZ, thus leading to inhibition of the FtsZ GTPase activity and its polymerization. Additionally, the crystal structure of CccR was determined, it was found to autoregulate its own expression using an inverted-repeat sequence located in its regulatory region. Moreover, CccR was shown to regulate gene expression also after it was delivered into neighboring kin cells. Thus, the authors demonstrated that CccR functions on competing cells (growth inhibition using its FIC domain) as well as on kin cells (transcriptional regulation using its HTH domain).

Major comments:

1. Fig. 3D and Fig. 3E, in both panels the effect of CccR on FtsZ was examined. In Fig. 3D the CccR effect on FtsZ GTPase activity and in Fig. 3E the CccR effect on FtsZ polymerization. In both assays, the concentration of CccR used was very high, and this does not fit the function of CccR as a protein harboring enzymatic activity – AMPylation. In the GTPase activity assay in order to reduce the FtsZ activity a ratio of 1:1 between the substrate (FtsZ) and enzyme (CccR) was required. The requirement for such a ratio is expected in the case where the binding of CccR to FtsZ per se results in the reduction of FtsZ GTPase activity but not if CccR enzymatic activity makes the effect. In the FtsZ polymerization assay, there was a need for 100microM of CccR in order to inhibit the polymerization of 10microM of FtsZ, this clearly does not fit the hypothesis that AMPylation of FtsZ results in the inhibition of polymerization, but it rather seems that the binding between the two proteins led to the observed inhibition. This issue should be resolved for example by using low concentration of CccR over a time course to examine its effect on FtsZ, or to use clean AMPylated-FtsZ in the two assays (without CccR).

Response: Thank you very much for your professional comments. Because both CccR and FtsZ are very unstable, which precluded us to use low concentration of CccR over a time course to examine its effect on FtsZ in **Fig. 3d** and **Fig. 3e** (**Respond Fig. 8**). Thus, to validate that it was AMPylation of FtsZ but not the binding between the two proteins led to the observed inhibition results, we included the catalytically inactive CccR^{H192A} mutant at the highest concentration of CccR protein as a control. The failure of the catalytically inactive CccR^{H192A} mutant to inhibit the GTPase activity and polymerization of FtsZ can support the conclusion that the AMPylation activity of CccR is essential for these effects.

Respond Fig. 8. CccR affects the GTPase activity and polymerization of FtsZ.

a, CccR treatment abolished the GTPase activity of FtsZ in a dose-dependent manner. The data are the mean \pm SD from three biological replicates. *P*-values from all data were determined using a two-sided, unpaired Student's *t*-test, and differences were considered significant at *P* < 0.05. ***, *P* < 0.001.

b, CccR inhibits FtsZ polymerization. Polymerized FtsZ treated by different concentrations of CccR was separated by ultracentrifugation. The resulting supernatants and pellets were analyzed by SDS-PAGE followed by Coomassie brilliant blue staining (Upper panel). Lower panel: Quantification of the percentage of polymerized FtsZ versus total FtsZ. The band intensity was quantified with Image J.

2. Fig. 5C (lines 339-344) – The authors generated an *E. coli* strain with the *cccR* gene containing a mutation in the palindromes of the regulatory element (*P_{cccRm}*) which drastically increased CccR expression. The protein band in Fig. 5C in the relevant lane is very strong. How such a strain can be alive? The strain produces high level of a potent toxin which AMPylates FtsZ, how can this strain replicate? this result does come together with the other information presented in the manuscript regarding CccR effect on *E. coli*.

Response: Thank you very much for these insightful and professional comments. The reason is that the toxicity of CccR is not sufficient to inhibit cell growth completely. Although expression of CccR inhibits the growth of *E. coli* cells, we can still purify His₆-tagged CccR proteins from IPTG induced *E. coli* cells as long as enough cells were collected. Similarly, the low expression levels of CccR in Fig. 5C were sufficient to be detected by Western Blotting.

3. The authors identified the regulatory element recognized by CccR, which is rather special (line 329). The RNA-seq analysis resulted in the identification of 447 genes which were up- or down regulated after CccR delivery. It is highly likely that very few (if any) of these genes harbor the CccR regulatory element, therefore it is not clear how CccR affects their expression. It might be that CccR function as a FIC toxin is responsible for the effect (induction of stress) and not its function as a transcriptional regulator. The RNA-seq analysis

should be repeated with a CccR mutated in the FIC domain to validate that the change in gene expression occurs due to CccR function as a transcriptional regulator.

Response: Thank you very much for these professional comments and advice. As suggested, we have validated the transcriptomic data using quantitative reverse transcriptase polymerase chain reaction (qRT-PCR) analysis on 16 representative genes by direct adding the catalytically inactive CccR^{H192A} protein to Δ cccR mutant cells (Respond Fig. 9, new Supplementary Fig. 11a). The results indicated that, at least for these genes, the change in gene expression occurs not due to the effect of FIC toxin induced stress.

Respond Fig. 9. Validation of RNA-seq data via qRT-PCR.

Validation of RNA-seq results via qRT-PCR. Sixteen representative genes were chosen to validate the RNA-seq data by qRT-PCR. CccR^{H192A} (200 ng/mL) was added to the Δ cccR culture when OD₆₀₀=0.5 and continue incubation at 26°C until OD₆₀₀=1.0. RNA was extracted to evaluate the transcription of genes by qRT-PCR. Equal volume of PBS buffer was served as the control. The fold change means multiple of CccR^{H192A} treated over CccR^{H192A} untreated samples. Error bars indicate the SD. White bars represent RNA-seq data.

However, although we have convincingly demonstrated that CccR acts as a transcriptional regulator to auto-repress its own expression not only in its producing cells but also in neighboring cells via direct DNA binding, we totally agree with you that the results about the globally regulation of 447 genes is not so convincing. The main problem is that we cannot identify putative CccR binding elements with high identities in the promoter regions of these regulated genes. So, it is not clear how CccR affects their expression. It might either be that CccR un-specifically recognize these promoters as a transcriptional regulator, or be that CccR function as a FIC toxin is responsible for the effect (induction of stress) for part of these genes. It seems impossible to reveal the global regulatory mechanism of CccR on these genes in this manuscript, but we will work on this topic continually in the future. In this manuscript we mainly focused on the specifically regulation of CccR on its own expression, and these results are sufficient to support the role of CccR in cell-to-cell communication.

We have discussed these issues in the discussion section as follow: “Previously, it was reported that a contact-dependent growth inhibition (CDI) system in *Burkholderia thailandensis* delivers BcpA, a DNase toxin, to induce changes in the differential expression of 841 genes in immune target cells. Although the molecular mechanism underlying this process, termed contact-dependent signaling (CDS), requires the catalytic activity of BcpA, the details remain unclear. In this study, we provide direct evidence of a role of CccR as a transcriptional regulator that regulates gene expression by directly binding DNA. RNA-seq analysis resulted in the identification of 447 genes that were up- or downregulated more than 1.2-fold after CccR delivery. However, because we cannot identify putative CccR binding elements in the promoter regions of these regulated genes, it is not clear how CccR globally affects their expression. CccR may nonspecifically recognize promoters as a transcriptional regulator, or may function as an FIC toxin that is responsible for the effect (induction of stress). It will be interesting to determine whether CccR can act as a transcriptional regulator in prey cells. However, given that toxic CccR will inhibit prey cell growth, and no CccR binding elements have been identified in the *E. coli* genome, transcriptional regulator activity of CccR in prey cells can be ignored like BcpA (Lines 477-493)”

4. It was not described in the manuscript if there is a known antitoxin for CccR. The FtsZ proteins of *E. coli* and *Yersinia pseudotuberculosis* are identical at the N-termini where CccR AMPylates FtsZ. Therefore, it is expected that *Yersinia pseudotuberculosis* will harbor an anti-toxin which inhibit CccR activity. Is there any information about the CccR antitoxin?

Response: Thank you for raising this important issue. Actually, the immunity mechanism of CccR seems very complex and the involved antitoxin needs to be investigated in the future. We have discussed this important issue from several points including the inhibitory α_{inh} , intrinsic or extrinsic factors to modulate the expulsion of the inhibitory α_{inh} , auto-repress of expression, and possible interacting T6SS components in the discussion section as follow: “While the activity of Class I FIC can be directly regulated by removing the interacting antitoxin, Class II and III FIC proteins may require complex intrinsic or extrinsic factors to modulate expulsion of the inhibitory α_{inh} . For example, the toxicity of some Class III FIC proteins is regulated by oligomerization, autoadenylylation, and metal ions. Although Class II FIC proteins account for 80% of total FIC proteins, nearly nothing is known about their biological functions and activity modulation. Here, we found that the Class II FIC protein CccR auto-represses its own expression by acting as a transcriptional regulator, providing a new perspective for understanding the immune mechanism of toxic proteins. However,

future work is needed to investigate the identity of the intrinsic and/or extrinsic factors that maintain the α_{inh} in an inhibitory state in self cells, and trigger expulsion of the α_{inh} to relieve autoinhibition in nonself cells. While each antibacterial T6SS effector is often co-expressed with a cognate immunity protein encoded in a bicistron to protect the producing cells and sister cells from intoxication, no immunity protein for CccR was identified in its neighboring ORFs. However, one can imagine that when CccR is produced, it may be bound by T6SS components (Hcp, PAAR or unknown chaperones) to prevent it from being toxic. Its repressor function could aim to prevent toxicity in cases where its binding to T6SS components is insufficient. The same is true of sister cells. In the presence of many prey cells, secreted CccR proteins will be preferentially internalized by prey cells to kill them. After the prey cells are killed, secreted CccR protein can serve as a signal molecule to inform the population that its production is unnecessary, because it is mainly internalized by sister cells in these conditions. This hypothesis is partially supported by the findings that CccR exhibited significant higher affinity to *E. coli* FhuA than to *Yptb* FhuA and enter *E. coli* cells more efficiently than *Yptb* cells **(Lines 494-519).**”

Relevant References:

- Engel P., Goepfert A., Stanger F. V., Harms A., Schmidt A., Schirmer T. and Dehio C. Adenylylation control by intra- or intermolecular active-site obstruction in Fic proteins. *Nature* (2012). 482 (7383): 107-110.
- Perera L. A., Preissler S., Zaccai N. R., Prevost S., Devos J. M., Haertlein M. and Ron D. Structures of a deAMPylation complex rationalise the switch between antagonistic catalytic activities of FICD. *Nat Commun* (2021). 12 (1): 5004.
- Veyron S., Oliva G., Rolando M., Buchrieser C., Peyroche G. and Cherfils J. A Ca²⁺-regulated deAMPylation switch in human and bacterial FIC proteins. *Nat Commun* (2019). 10 (1): 1142.

Reviewer #1 (Remarks to the Author):

The authors have answered many of my concerns, greatly improving an already excellent work. I have no doubt that these concerns will be answered in future work beyond the scope of this manuscript.

Reviewer #2 (Remarks to the Author):

The authors have adequately addressed my comments, I have no additional concerns for this exciting manuscript. It is now suitable for publication in Nature Communications.

Reviewer #3 (Remarks to the Author):

I would like to commend the authors on doing such a thorough and extensive work to address my comments and concerns. Everything has been fully addressed and I would be happy to recommend publication. Congratulations on a beautiful paper!